# Estimating agricultural ammonia volatilization over Europe using satellite observations and simulation data

Rimal Abeed[1], Camille Viatte[1], William C. Porter[2], Nikolaos Evangeliou[3], Cathy Clerbaux[1,4], Lieven Clarisse[4], Martin Van Damme[4,5], Pierre-François Coheur[4], and Sarah Safieddine[1]

[1]LATMOS/IPSL, Sorbonne Université, UVSQ, CNRS, Paris, France
[2]Department of Environmental Sciences, University of California, Riverside, CA 92521, USA
[3]Norwegian Institute for Air Research (NILU), Department of Atmospheric and Climate Research (ATMOS), Kjeller, Norway
[4]Université libre de Bruxelles (ULB), Spectroscopy, Quantum Chemistry and Atmospheric Remote Sensing (SQUARES), Brussels, Belgium
[5]Belgian Institute for Space Aeronomy (BIRA-IASB), Brussels 1180, Belgium

*Correspondence to*: Rimal Abeed rimal.abeed@latmos.ipsl.fr

**Abstract**

Ammonia ($NH_3$) is one of the most important gases emitted from agricultural practices. It affects air quality and the overall climate, and in turn influenced by long term climate trends as well as by short term fluctuations in local and regional meteorology. Previous studies have established the capability of the Infrared Atmospheric Sounding Interferometer (IASI) series of instruments aboard the Metop satellites to measure ammonia from space since 2007. In this study, we explore the interactions between atmospheric ammonia, land and meteorological variability, and long-term climate trends in Europe. We investigate the emission potential ($\Gamma_{soil}$) of ammonia from the soil, which describes the soil – atmosphere ammonia exchange. $\Gamma_{soil}$ is generally calculated in-field or in laboratory experiments; here, and for the first time, we investigate a method which assesses it remotely using satellite data, reanalysis data products, and model simulations.

We focus on ammonia emission potential during March 2011, which marks the start of growing season in Europe. Our results show that $\Gamma_{soil}$ ranges from $2 \times 10^3$ to $9.5 \times 10^4$ (dimensionless) in a fertilized cropland, such as in the North European Plain, and is of the order of $10 – 10^2$ in a non-fertilized soil (e.g. forest and grassland). These results agree with in-field measurements from the literature, suggesting that our method can be used in other seasons and regions in the world. However, some improvements are needed in the determination of mass transfer coefficient $k$ (m $s^{-1}$), which is a crucial parameter to derive $\Gamma_{soil}$.

Using a climate model, we estimate the expected increase in ammonia columns by the end of the century based on the increase in skin temperature (T skin), under two different climate scenarios. Ammonia columns are projected to increase by up to 50 %, particularly in Eastern Europe, under the SSP2-4.5 scenario, and might even double (increase of 100 %) under the SSP5-8.5 scenario. The increase in skin temperature is responsible for a formation of new hotspots of ammonia in Belarus, Ukraine, Hungary, Moldova, parts of Romania, and Switzerland.

## 1. Introduction

Ammonia ($NH_3$) emissions have been increasing in a continuous manner from 1970 to 2017 (McDuffie et al., 2020). During the period 2008 – 2018 alone, the increase in ammonia columns in Western and Southern Europe accounted to 20.8 % $yr^{-1}$ (± 4.3 %), and to 12.8 (± 1.3 %) globally (Van Damme et al., 2021). Although ammonia alone is stable against heat and light, it is considered a very reactive base, whereas it constitutes the largest portion of the reactive nitrogen ($N_r$) on Earth. The vast majority of atmospheric ammonia not deposited is transformed into fine particulate matter ($PM_{2.5}$) composed of ammonium ($NH_4^+$), through acid – base chemical reactions with available acids in the environment, namely sulfuric acid ($H_2SO_4$), hydrochloric acid (HCl), and nitric acid ($HNO_3$) (Yu et al., 2018), while only 10 % of the total ammonia gas are believed to be oxidized by hydroxyl radicals ($OH^-$) (Roelle and Aneja, 2005). $PM_{2.5}$ has degrading effects on human health, especially respiratory diseases (Bauer et al., 2016). In addition to agriculture, ammonia can be emitted from industrial processes, biomass burning (Van Damme et al., 2018), and natural activities such as from seal colonies (Theobald et al., 2006).

Soils are known to be a source of atmospheric ammonia, especially in areas of intensive agricultural practices (Schlesinger and Hartley, 1992), and this is due to enriching the soil with the reactive nitrogen present in fertilizers. The increase in the application of synthetic fertilizers, and intensification of agricultural practices is believed to be the dominant factor of the global increase in ammonia emissions over the past century (Behera et al., 2013; McDuffie et al., 2020).

Following the application of fertilizers, ammonium and ammonia are released in the soil. Prior to its volatilization, ammonia in the soil exists either in the gas phase ($NH_{3\,(g)}$) or in the aqueous/liquid phase ($NH_{3\,(aq)}$), the equilibrium between both states of ammonia is governed by Henry's law (Wentworth et al., 2014), as shown in A. The dissociation of ammonia in soil water is affected by soil acidity (pH) and temperature (Roelle and Aneja, 2005) (Eq. (A-1) and (A-2) in Appendix A); it is explained by the dissociation constant $K_{NH_4^+}$. Once released to the atmosphere, ammonia near the surface exists is in the gas phase, hence Henry's law describes the equilibrium between ammonia in the soil (liquid phase), and near the surface (gas phase). This bi-directional exchange between the soil and the atmosphere will continue until the equilibrium is reached, and this occurs when ammonia concentration is equal to the compensation point $\chi_{NH_3}$ (the concentration of ammonia at equilibrium). The flux of ammonia from the soil to the atmosphere (emission) occurs when the concentration of atmospheric ammonia is less than the compensation point $\chi_{NH_3}$, while ammonia deposition occurs when the concentration of ammonia is equal to or greater than $\chi_{NH_3}$ (Flechard et al., 2011; Wichink Kruit, 2010). It is then crucial to quantify the compensation point in order to understand this bi-directional exchange. The main variables needed to calculate $\chi_{NH_3}$ are soil temperature (T skin) and $\Gamma_{soil}$, which is a dimensionless ratio between ammonium and pH ($NH_4^+{}_{(aq)}$ and $H^+{}_{(aq)}$ concentrations, respectively, in the soil). All the equations are described in Appendix A (Eq. (A-1) to (A-15)).

The soil emission potential ($\Gamma_{soil}$) has been thoroughly investigated in field and controlled laboratory environments (e.g. David et al., 2009; Flechard et al., 2013; Massad et al., 2010; Mattsson et al., 2008; Nemitz et al., 2000; Wentworth et al., 2014, among others). $\Gamma_{soil}$ is dimensionless and it can range from 20 (non-fertilized soil in a forest) to the order of $10^6$ (mixture of slurry in a cropland). It is found to peak right after fertilizers application, due to the increase in ammonium content in the soil (a product of urea hydrolysis), reaching pre-fertilization levels 10 days following the application (Flechard et al., 2010; Massad et al., 2010). Little information exists on regional or global scales to assess the large-scale spatial variability of ammonia emission potentials.

In order to meet the needs for a growing population, agricultural practices have intensified during the period 2003 – 2019 (more fertilizer use per surface area), resulting in an increase in the net primary production (NPP) per capita

(Potapov et al., 2022), subsequently increasing volatilized ammonia (increase in nitrogen soil content, and cultivated lands). In Europe alone, the area of croplands increased by 9 % from 2003 to 2019, and most of the expansion took place on lands that were abandoned for more than 4 years (Potapov et al., 2022). Between the year 2008 and 2018, the increase in atmospheric ammonia columns accounted to 20.8 % (± 4.3 %) in Western and Southern Europe (Van Damme et al., 2021). With the increase in croplands area and agricultural activities, climate change will have a significant effect on agricultural practices, with warmer climates enhancing the volatilization of ammonia from soils, especially in intensely fertilized lands (Shen et al., 2020).

This study aims at exploring ammonia emission potential/volatilization in Europe, using infrared satellite data of ammonia columns, reanalysis temperature data, and chemical transport model simulations to provide information on chemical sources and sinks. We specifically study the relationship between satellite-derived ammonia concentration at the start of the growing season, soil emission potentials and their spatial variability over Europe during March of 2011. Section 2 provides the methods/datasets used. Simulation results are described in Sect. 3, and regional emission potentials are shown and discussed in Sect. 4. Using a climate model, future projections of ammonia columns are investigated under different climate scenarios in Sect. 5. Discussion and conclusions are listed in Sect. 6.

## 2. Methodology

### 2.1. Calculation of the emission potential

In this study, we use IASI satellite data to calculate the ammonia emission potential $\Gamma_{soil}$ instead of field soil measurements. In field studies, $\Gamma_{soil}$ is calculated by measuring the concentration of ammonium ($NH_4^+$) and $H^+$ ($10^{-pH}$) in the soil; the ratio between both of these concentrations is $\Gamma_{soil}$. In this study, we use infrared satellite ammonia to have a regional coverage over Europe. With these, we cannot monitor soil content of ammonium nor its pH. This renders the remote $\Gamma_{soil}$ calculation challenging, and less straight forward. The full derivation of the equation used to calculate the emission potential is explained in Appendix A, and was briefly described in the introduction. In short, upon its dissolution in the soil water, ammonia follows Henry's law. In steady state conditions between the soil and the near surface, the amount of the ammonia emitted and lost is considered equal. Based on this assumption, the soil emission potential (dimensionless) is calculated as follows Eq. (2-1) or Eq. (A-15) in Appendix A:

$$\Gamma_{soil} = \frac{[NH_3]^{col} \,.\, T_{soil}}{\exp(\frac{-b}{T_{soil}})} \frac{M_{NH_3}}{a \,.\, N_a \,.\, c'} \cdot \frac{1}{k\tau} \qquad \textbf{(2-1)}$$

Where $[NH_3]^{col}$ is the total column concentration of ammonia (molecules cm$^{-2}$), measured by satellite remote sensors, $T_{soil}$ is the soil temperature at the surface, which can be expressed as the skin temperature, T skin (Kelvin), $a$ and $b$ are constants ($a = 2.75 \times 10^3\ g\ K\ cm^{-3},\ b = 1.04 \times 10^4\ K$), $M_{NH_3}$ is the molar mass of ammonia gas ($M = 17.031\ g\ mol^{-1}$), and $N_a$ is Avogadro's number ($N_a = 6.0221409 \times 10^{23}\ molecules\ mol^{-1}$), $c'$ is equals to 100 and is added to convert $k$ from m s$^{-1}$ to cm s$^{-1}$ (since $[NH_3]^{col}$ is in molecules cm$^{-1}$), and $\tau$ the lifetime of ammonia (seconds).
$k$ is the soil – atmosphere exchange coefficient or deposition velocity (cm s$^{-1}$), also known as the mass transfer coefficient (this nomenclature will be used in this study). It is found to be affected by the roughness length of the surface, wind speed, the boundary layer height (Olesen and Sommer, 1993; Van Der Molen et al., 1990), and pH

(Lee et al., 2020). It can be explained by a resistance model often used to explain the exchange between the surface and the atmosphere (Wentworth et al., 2014). Different studies provide look up tables values of $k$ for different land cover types and different seasons based on this resistance model (Aneja et al., 1986; Erisman et al., 1994; Phillips et al., 2004; Roelle and Aneja, 2005; Svensson and Ferm, 1993; Wesely, 1989).
In general, the mass transfer coefficient $k$ is in the order of $10^{-3}$ to $10^{-2}$ m $s^{-1}$ in a mixture of soil and manure, and $10^{-6}$ to $10^{-5}$ m $s^{-1}$ in a mixture of manure alone (Roelle and Aneja, 2005). We discuss and provide more information on $k$ in Sect. 4, and additional details on this calculation in general are provided in Appendix A**.**

### 2.2. IASI ammonia, ERA5 T skin, and MODIS Land cover

The Infrared Atmospheric Sounding Interferometer (IASI) is considered advanced the most innovative instrument onboard the polar-orbiting Metop satellites (Klaes, 2018). Three IASI instruments are onboard Metop-A, B and C, the series of satellites launched by the EUMETSAT (European Organization for the Exploitation of Meteorological Satellites) in 2006, 2012, and 2018, respectively. The Metop-A satellite was de-orbited in October 2021 (Lentze, 2021), and as a result only two instruments (IASI-B and C onboard Metop-B and C) are operating today. The observations from IASI cover any location on Earth at 9:30 in the morning (AM) and in the evening (PM), local solar time. It can detect a variety of atmospheric species including trace gases (Clerbaux et al., 2009). The IASI Fourier-transform spectrometer monitors the atmosphere in the spectral range between 645 and 2760 $cm^{-1}$ (thermal infrared), and is nadir-looking. IASI has a swath width that measures 2200 km, with a pixel size of ~12 km.

Ammonia was first detected with IASI using the $\upsilon_2$ vibrational band of ammonia (~950 $cm^{-1}$) (Clerbaux et al., 2009; Coheur et al., 2009). The ammonia total columns used in this study are the product of an Artificial Neural Network and re-analyzed temperature data from the European Centre for Medium-Range Weather Forecasts (ECMWF) product ERA5 ANNI-$NH_3$-v3R-ERA5 (Van Damme et al., 2021). Several studies used ammonia data from IASI to study hotspots of ammonia of different source types including both natural and anthropogenic sources (Clarisse, Van Damme, Clerbaux, et al., 2019; Clarisse, Van Damme, Gardner, et al., 2019; Dammers et al., 2019; Van Damme et al., 2018, 2021; Viatte et al., 2021). Recently, IASI observations were used to study the effect of war and conflict on agricultural practices in Syria (Abeed et al., 2021).

Fewer errors on the retrieval were observed during the day and over land (Van Damme et al., 2017), hence, we use only daytime ammonia measurements from IASI. Comparisons with ammonia measured using a ground-based instrument showed a good correlation of R=0.75 (Viatte et al., 2021). Satellite ammonia data from CrIS (Crosstrack Infrared Sounder) (Shephard and Cady-Pereira, 2015) were compared with those from IASI, and were equally found to give similar results when looking at concentrations from a wildfire (Adams et al., 2019), showing consistency when studying seasonal and inter-annual variability (Viatte et al., 2020).

In addition to ammonia, we look at skin temperature (T skin or land surface temperature LST) data from the ECMWF reanalysis (ERA5) at a grid resolution of 0.25 × 0.25 ° (Hersbach et al., 2020). ERA5 Temperatures are interpolated temporally and spatially to the IASI morning overpass (~9:30 A.M. local time), since we only consider daytime ammonia. ERA5 temperature data are also used in the retrieval process of the ammonia data we used in this study $NH_3$-v3R-ERA5 (Van Damme et al., 2021). T skin is defined as the temperature of the uppermost surface layer when radiative equilibrium is reached. It also represents the theoretical temperature required in order to reach the surface energy balance (ECMWF, 2016).

In order to assign a mass transfer coefficient $k$ to each land type, the moderate resolution imaging spectroradiometer (MODIS), a series of instruments orbiting the Earth aboard the Aqua and Terra satellites, is used. The data product MCD12Q1 (version 6) is a combined Aqua/Terra Land cover type product, with a spatial resolution of 500 m. This product provides maps of land cover type from 2001 through 2019 (Sulla-Menashe and Friedl, 2018). From the land use categories included in the MOD12Q1 product (Belward et al., 1999) we focus on croplands, forests, shrublands, and grasslands. We do not include barelands, snow cover, and urban areas in our analysis; we are not interested in studying these surfaces, since we focus on ammonia volatilization from the soil in areas amended with fertilizers. We show the emission potential in Forests and grasslands/shrublands for comparison with values in the literature. In an attempt to calculate an emission potential (Eq. (2-1)) that is relevant to the land cover/use, we therefore assign a mass transfer coefficient $k$ to each land type based on literature values (Aneja et al., 1986; Erisman et al., 1994; Roelle and Aneja, 2005; Svensson and Ferm, 1993; Wesely, 1989) and we discuss it in Sect. 4.

## 2.3. Model simulations

### 2.3.1. GEOS-Chem Chemistry Transport Model

In this study we use version 12.7.2 of the GEOS-Chem chemical transport model (Bey et al., 2001). The model is driven by the Modern-Era Retrospective Analysis for Research and Applications version 2 (MERRA-2) reanalysis product, including nested domains over Europe at a 0.5° × 0.625° horizontal resolution. MERRA-2 is the second version of the MERRA atmospheric reanalysis product by NASA Global Modulation Assimilation Office (NASA/GMAO) (Gelaro et al., 2017). Boundary conditions for the nested domains are created using a global simulation for the same months at 2° × 2.5° resolution. We generate model output for March of 2011, preceded by a one month of discarded model spin-up time for the nested run, and two months for the global simulation. March corresponds well to the month of fertilizer application in Europe, and as such to the beginning of the growing season (FAO, 2022; USDA, 2022).

Output includes the monthly mean for selected diagnostics. Anthropogenic emissions are taken primarily from the global Community Emissions Data System (CEDS) inventory (Hoesly et al., 2018). Biogenic non-agricultural ammonia, as well as ocean ammonia sources, are taken from the Global Emission Inventories Activities database (GEIA, (Bouwman et al., 1997)). Open fire emissions are generated using the GFED 4.1s inventory (Randerson et al., 2015). We used the Harmonized Emissions Component module (HEMCO) to obtain the ammonia emissions over Europe (Keller et al., 2014).

### 2.3.2. EC-Earth Climate model

To analyze how future climate will affect ammonia concentration and emission potential, we use the ECMWF climate model the European Earth Consortium climate model (EC-Earth, http://www.ec-earth.org/). While other climate models exist, we choose this one because the ammonia product from IASI uses ERA5 for the retrievals and we calculate the emission potential from the T skin product of ERA5. The reanalysis uses the ECMWF Integrated Forecasting System for the atmosphere–land component (IFS). IFS is also used in EC-Earth and is complemented with other model components to simulate the full range of Earth system interactions that are relevant to climate (Döscher et al., 2021). We note that the versions of the IFS models used in ERA5 and in EC-Earth are not identical as the climate model product is not assimilated and is not initialized with observations

several times a day like ERA5. The EC-Earth simulations are included in the Climate model intercomparison project, phase 6 (Eyring et al., 2016), part of the Intergovernmental Panel on Climate Change (IPCC) report of 2021 (Masson-Delmotte, et al., 2021). We use the so-called Scenario Model Intercomparison Project (ScenarioMIP), covering the period [2015 – 2100] for future projections under different shared socio-economic pathways (SSP) (Riahi et al., 2017). We analyze two scenarios, the SSP2-4.5 corresponding to "middle of the road" socio-economic family with a nominal 4.5W/m$^2$ radiative forcing level by 2100 - approximately corresponding to the RCP-4.5 scenario, and the SSP5-8.5 marks the upper edge of the SSP scenario spectrum with a high reference scenario in a high fossil-fuel development world throughout the 21st century.

## 3. GEOS-Chem model simulation: validation and analysis

### 3.1. GEOS-Chem validation with IASI

In order to analyse how well the model simulates atmospheric ammonia, we use the simulated GEOS-Chem monthly averaged (March 2011) ammonia total columns output (Sect. 2.3.1). We compare those to the IASI total columns of ammonia gridded on the same horizontal resolution (0.5° × 0.625°) and over the same month.

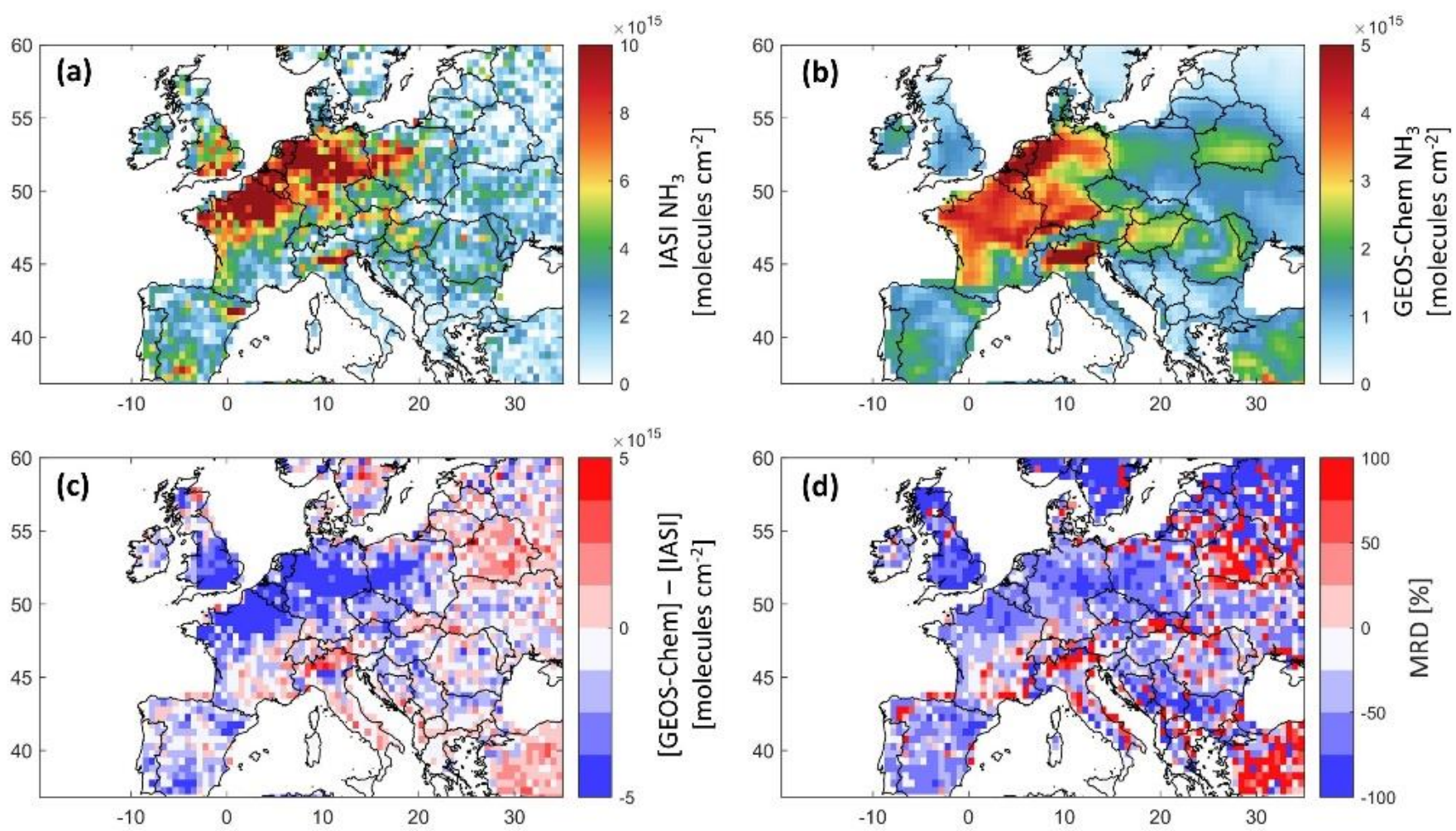

**Figure 1. Ammonia total column concentrations from IASI (panel a), and GEOS-Chem (panel b), the difference between both datasets (panel c) in molecules cm$^{-2}$, and the Mean Relative Difference (MRD) in % (panel d); all data are a monthly average of March 2011, and over Europe at a 0.5° × 0.625° grid resolution. Note that the colour bar limits are different between panels (a) and (b).**

Figure 1 shows the IASI $NH_3$ distribution (Figure 1a), and that from GEOS-Chem (Figure 1b), the bias between the two (Figure 1c), and the mean relative difference MRD (Figure 1d), all during March 2011. MRD is calculated as the mean of the ratio $\frac{(GeosChem\ NH_3 - IASI\ NH_3) \times 100}{IASI\ NH_3}$ at each grid point.

Generally, both GEOS-Chem and IASI show coincident sources of ammonia, reflecting the good ability of the model to reproduce ammonia columns over major agricultural source regions in Europe. The bias between IASI and GEOS-Chem and the MRD are shown in Figure 1c and d. Ammonia columns from GEOS-Chem are underestimated by up to $2 \times 10^{16}$ molecules/cm² in some source regions/over hotspots, especially in England, North Eastern France, the North European Plain (Netherlands, Belgium), and Spain (around Barcelona). Similar results were found in the study of Whitburn et al. (2016), in which they show that GEOS-Chem underestimates ammonia columns by up to $1 \times 10^{16}$ molecules/cm² in Europe on a yearly average in 2009, notably in the North European Plain. It is important to note that, in our study, we compare only one month of data (March, 2011) that marks the start of the growing season in the majority of the countries of interest (FAO, 2022; USDA, 2022). The differences are mainly because of the time coincidence, and the fact that only cloud-free data are used to retrieve ammonia; IASI observes ammonia during the satellite overpass (~9:30 AM local time), whereas the GEOS-Chem simulation is averaged over the whole month including all hours of the day. In Western and Northern Europe, the MRD is mostly less than −50 %, for instance, in the North European Plain (−49 %). If we look at the average MRD in regions of focus, we see that the Po Valley in Italy has the highest MRD value (+110 %), whereas the best represented region is New Aquitaine in the southwest of France (−20 %). The rest of the regions have mean MRDs that fluctuate between −64 % and – 42 %. A summary of the results of this study, including the MRD over some source regions is listed in Table 1. Although the bias and MRD can be considered high, the spatial distribution is consistent between IASI and GEOS-Chem. Therefore, according to the steady state approximation, the meteorological and soil parameters affecting one dataset (e.g. IASI $NH_3$) are applicable to the other (e.g. model simulation). It is worth noting that although we do not use the latest version of GEOS-Chem, the results we obtain reflects our current understanding of the regional chemistry at this horizontal and temporal resolution.

### 3.2. Ammonia emissions, losses and lifetime in Europe

In order to understand the $NH_3$ spatial variability in Europe during the application of fertilizers, a detailed analysis of the output of the GEOS-Chem simulation for the month of March 2011 is shown in Figure 2.
The anthropogenic sources (i.e. mainly agriculture) contribute 83 % of the total ammonia emissions during March 2011 in Europe. The ammonia emissions from natural sources (i.e. soil of natural vegetation, oceans, and wild animals) follow representing 16 % of the total emissions, whereas the remaining 1 % correspond to the ammonia emissions from biomass burning and ships (not shown here).
Figure 2a shows ammonia monthly emissions. Most of them are due to agricultural activities (not shown here); we identify 8 source regions which we investigate thoroughly in this study shown as rectangles A to H. The highest agricultural sources over Europe include the North European Plain, Brittany, and the Po Valley (regions C, D, and F).
In the calculation of the total loss of ammonia (Figure 2b), we considered dry deposition, chemistry, transport, and wet deposition (in which we included ammonia loss to convection) from the GEOS-Chem model simulation, which are all possible loss processes for ammonia (David et al., 2009). Figure 2b shows that the largest losses occur logically where we have the highest sources detected (see Figure 2a).

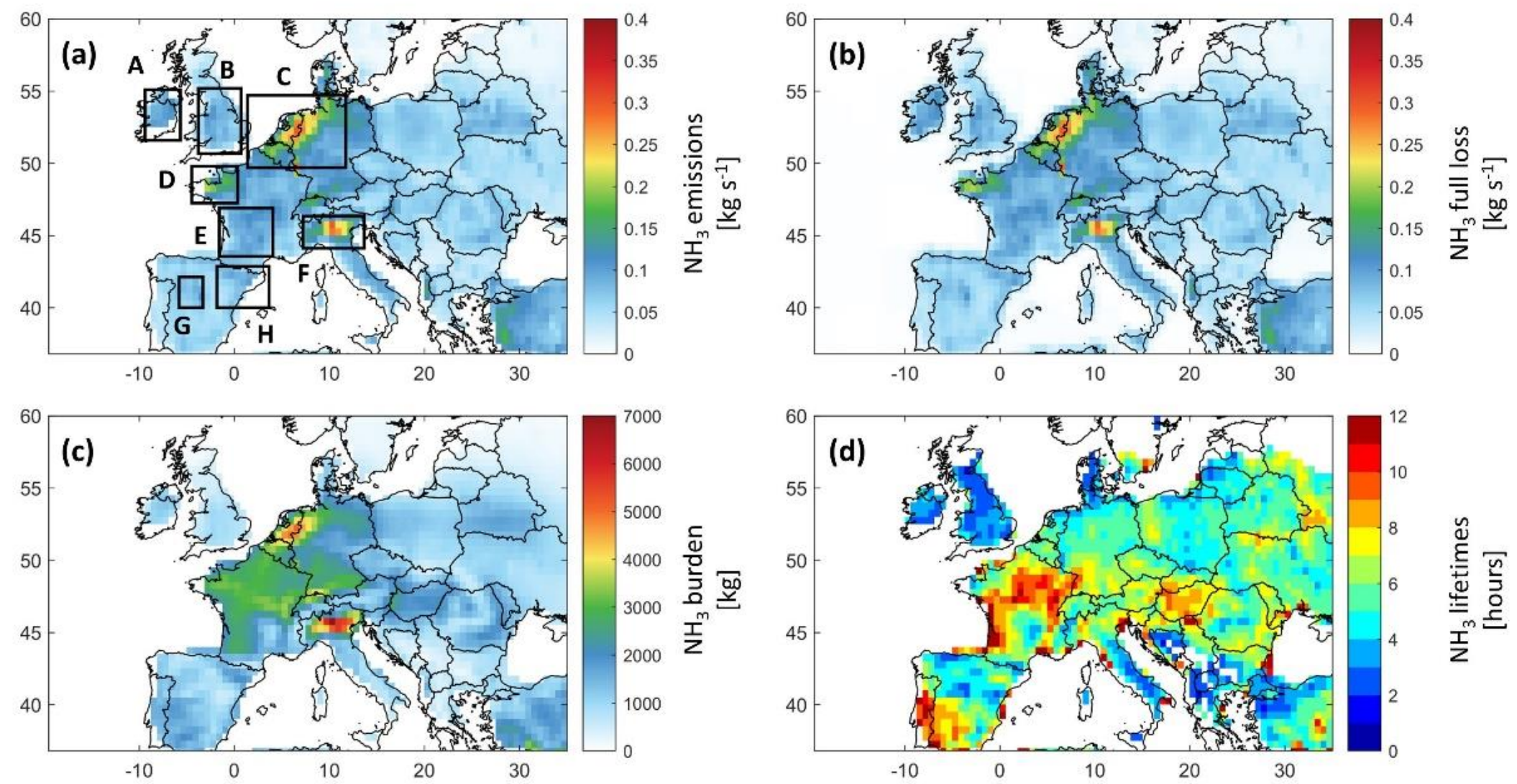

**Figure 2. Ammonia budget in Europe from GEOS-Chem: (a) Ammonia emissions from the Harmonized Emissions Component module (HEMCO) in kg s$^{-1}$ with our regions of interest shown in rectangles, (b) ammonia full loss in kg s$^{-1}$, (c) ammonia total burden in kg, and (d) ammonia lifetime in hours. All plots refer to March 2011 and are presented at a 0.5° × 0.625° grid resolution.**

The total ammonia burden (Figure 2c) is calculated as the integrated sum of all ammonia columns in the model grid box. We can clearly detect ammonia hotspots over Europe, in particular the North European Plain, Brittany and the Po Valley, all regions characterized by intense agricultural activities, as the total emissions and deposition show (Figure 1 and Figure 2). We also see that the burden is generally the highest over France, Belgium, The Netherlands, and parts of Germany and Italy.

We finally get the lifetime $\tau_{ss}$ of ammonia (Figure 2d). In the case of a gas with a short lifetime, such as ammonia, the emissions are relatively well-balanced spatially by eventual sinks/losses (steady-state approximation). Therefore, we can calculate a steady-state lifetime as the ratio between the total burden $B$ (Figure 2c) and the total emissions $E$ or losses $L$ (sum of all emitted / lost molecules, Figure 2a or b) using the following equation: $\tau_{ss} = {}^{B}/_{L}$ (Plumb and Stolarski, 2013).

We note that the $\tau_{ss}$ is more or less the same whether we calculate it using the losses or the emissions. For instance, in selected source regions (rectangles in Figure 2a) the total emissions and losses are very close with very low biases that are less than 2% (not shown here). Our results show that $\tau_{ss}$, on a monthly average, can go up to 12 hours, and it can reach 1 day (24 hours) in coastal regions such as region E in New Aquitaine in France. The latter can be related to the high probability of air stagnation is in that area in comparison to Northern Europe (Garrido-Perez et al., 2018), since higher $PM_{2.5}$ pollution episodes were found under stagnant meteorological conditions (AQEG, 2012); and ammonium molecules carried on these $PM_{2.5}$ can transform back into ammonia. Our results agree with the literature suggesting a residence time between a few hours to a few days (Behera et al., 2013; Pinder et al., 2008), and with those calculated by Evangeliou et al. (2021) over Europe, showing a monthly average of ammonia lifetime that ranges from 10 to 13 hours in Europe. The figure adapted from Evangeliou et al. (2021) is

shown in supplementary material (Figure S1). Shorter lifetimes from industrial sources of ammonia were reported in Dammers et al. (2019), with a mean lifetime of ammonia that is equal to 2.35 hours (±1.16). A recent study found lifetimes of ammonia that vary between 5 and 25 hours, roughly, in Europe (Luo et al., 2022); these values are higher since, in addition to ammonia loss, Luo et al. (2022) included the loss of ammonium, and thus considering the loss of ammonia only terminal when the ammonium is also lost/deposited. This approach is not considered here nor in Evangeliou et al. (2021).

Notably, ammonia lifetime and burden (Figure 2c, and d) each have different spatial distribution compared to the other 2 panels (Figure 2a, and b). The ammonia residence time in the atmosphere varies depending on the sources and more importantly on the locally dominant loss mechanisms. For this reason, in Figure 3, we show the relative contribution of the ammonia loss mechanisms, presented as pie charts, for the agricultural source regions shown in black boxes in Figure 2a.

The fastest loss mechanisms are either chemical (i.e. in the vast majority transformation to particulate matter) or through wet and dry deposition (Tournadre et al., 2020). Figure 3 shows that more than 50 % of the ammonia molecules in the atmosphere are lost to chemical reactions in most of the regions (A, B, C, H, and F). The shortest residence time of ammonia is observed in England, where the chemical removal was significantly higher than other sinks and represented up to 73 % of the total ammonia loss pathways, suggesting a rapid transformation into inorganic particulate matter ($PM_{2.5}$). In the regions D, G and E the chemical loss makes up 50 %, 49 %, and 42 %, respectively. In fact, in March 2011, PM was found to be mostly composed of inorganic nitrate (41 %), and ammonium (20 %) (Viatte et al., 2022) over Europe, both of which are products of atmospheric ammonia. Nitrate-bearing $PM_{2.5}$ are formed when nitric acid ($HNO_3$) reacts with ammonia (Yang et al., 2022), and ammonium is a direct product of the hydrolysis of ammonia. 41% of the nitric acid formed in the atmosphere is produced from the reaction between nitrogen dioxide ($NO_2$) and the hydroxyl radical (OH) (Alexander et al., 2020). These chemical pathways help explain the large chemical losses in most of the regions studied in Figure 3.

Ammonia loss to transport is the highest in regions neighboring the Atlantic Ocean, accounting for 30 %, 27 %, 32 %, and 34 % of total sinks in regions A, D, E, and G respectively. These regions are exposed to the North Atlantic Drift, also known as the Gulf Stream, that is associated with high wind speed and cyclonic activity (Barnes et al., 2022). In other regions, 14 % to 22 % of the total ammonia is lost to transport mechanisms, and in all regions, 11 to 22 % is lost to dry deposition (Figure 3). During March, precipitation is relatively lower as compared to winter (December, January, February) in Europe. Furthermore, 2011 was a particular dry year compared to the 1981 – 2010 average (Met Office, 2016). Drought was reported to be severe in areas such as France, Belgium and the Netherlands, and moderate in England and Ireland (EDO, 2011). This can help explain the low percentage of wet deposition during March 2011 (1 to 5 % out of the total loss of ammonia).

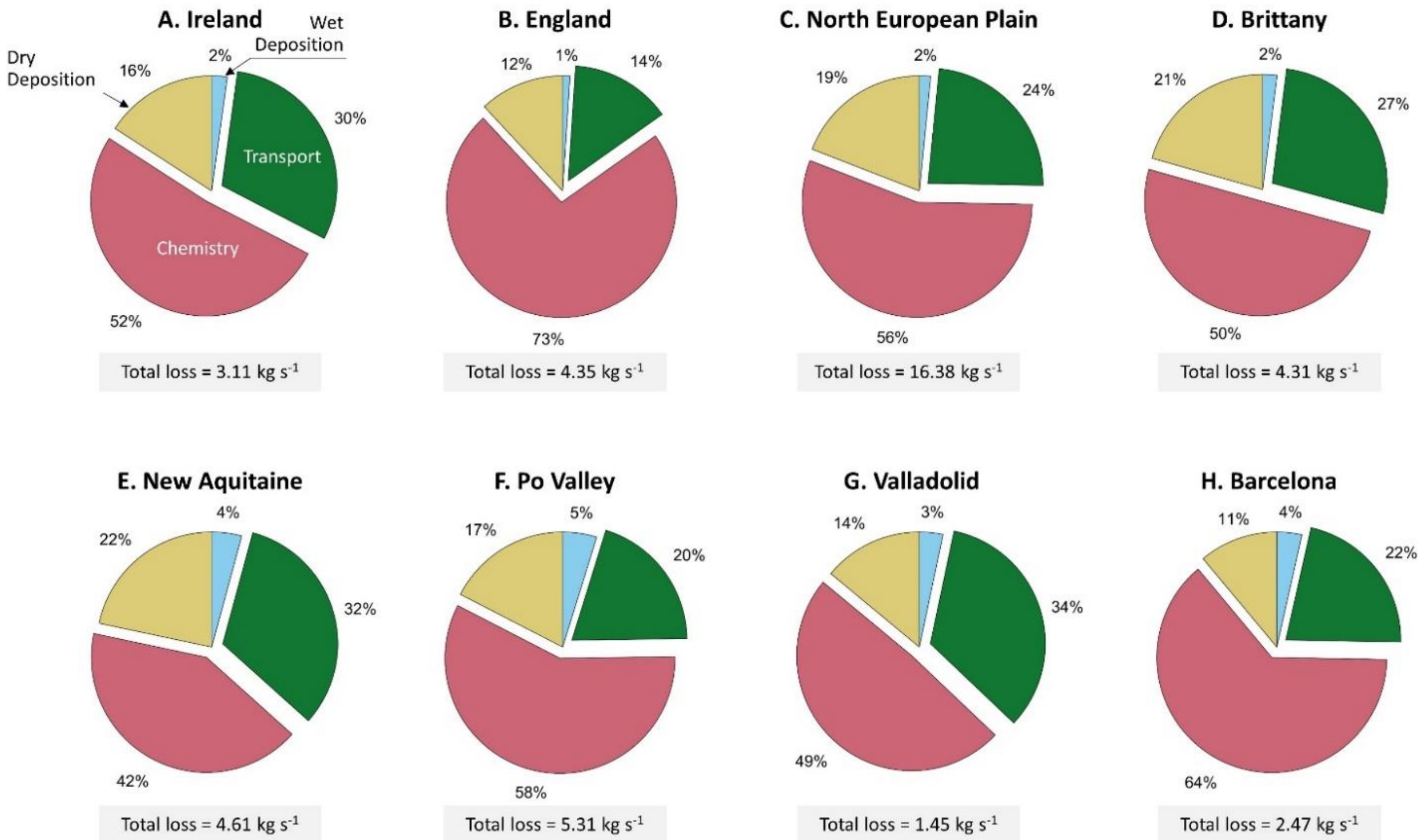

**Figure 3. Repartition of the ammonia loss mechanisms for major agricultural areas in Europe, during March 2011, as retrieved from GEOS-Chem, with the total ammonia loss shown in a grey box under each pie chart (kg $s^{-1}$). The regions are shown in black boxes in Figure 2a.**

## 4. Ammonia emission potential over Europe

To calculate emission potential, a calculation of the mass transfer coefficient $k$, which relates to the land type, is necessary. Figure 4 shows the land cover type from MODIS in Europe (left panel), and the corresponding assigned mass transfer coefficient $k$ (right panel) needed to calculate the emission potential (Eq. (2-1)). In order to choose a mass transfer coefficient that is convenient for the different land types relevant to this study, we searched for $k$ values in the literature. Not all land types have been studied for ammonia transfer coefficient.

For water bodies and other land types that are not considered here (see Sect. 2.2), the mass transfer values $k$ were set to zero and represented in grey colour in Figure 4. In a laboratory experiment, Svensson et al. (1993) reported $k = 4.3 \times 10^{-3}$ m $s^{-1}$ for a mixture of soil and swine manure, as therefore, we assign this value to croplands. Due to the lack of $k$ values for non-fertilized forests, shrublands and grasslands in the literature, we used values originally assigned for $SO_2$, bearing in mind that these are approximate values and they reflect mostly the conditions of the soil cover type (short, medium or tall grass). To assign a $k$ value for forests, we used values reported in Aneja (1986) ($k = 2 \times 10^{-2}$ m $s^{-1}$), which originally represent deposition velocity (mass transfer) of

$SO_2$ in a forest (high crops). For shrublands and grasslands (the two land types have the same $k$), we used the value $k = 8 \times 10^{-3}$ m s$^{-1}$ that has been reported in Aneja et al. (1986) as the deposition velocity (mass transfer) of $SO_2$ in a grassland (medium crops). These values are the best attempt to test the validity of using MODIS and lookup tables of $k$ values to calculate a realistic soil emission potential. As a result, Figure 4 (left panel) includes 5 land types, while k values are reported for 4 land types (other land type/water, croplands, forests, and shrublands/grasslands).

After choosing the $k$ values, we assigned them for each land type on the (500 m × 500 m) grid. We then extrapolate the array with the $k$ values from 500 m × 500 m to the resolution of GEOS-Chem (0.5° × 0.625° grid box). This leads to averaging different fine pixels with different land cover types into a coarser grid. The result is shown on the right panel of Figure 4.

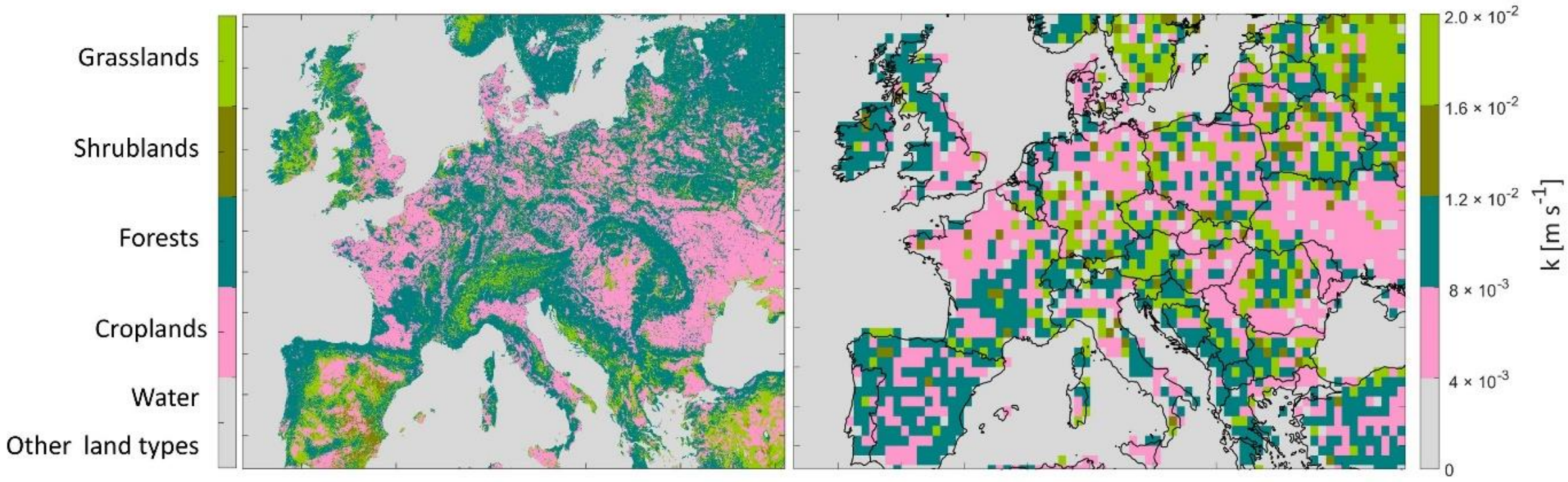

**Figure 4. MODIS Land Cover Type, at a 500 m × 500 m grid box (left panel), and interpolated mass transfer coefficient *k* on a horizontal resolution of 0.5° × 0.625° grid box (right panel).**

Uncertainties of this methodological approach can be summarized as follows:

(1) The $k$ value assigned for croplands is approximate and therefore not the same in every cropland over Europe.
(2) The $k$ value assigned for forests represents the $SO_2$ exchange in high croplands, and ammonia might change especially when the latter is highly affected by humidity; it can easily dissolve in the water film on leaves in high humid conditions.
(3) The extrapolation of a fine array (500 m × 500 m) will merge several grids together and average them in order to construct the coarser grid box (0.5° × 0.625°); the result is therefore an average that might mix croplands with neighboring forests/barelands/grasslands. This leads to a range of different $k$ values that are shown on Figure 4.

Using the land-type specific $k$ value is necessary in order to reflect realistic emissions potential, as ammonia exchange in a forest is different from that of croplands or unfertilized grasslands, due to different barriers (long, medium or short crop / grass), and ammonium soil content in each land type.

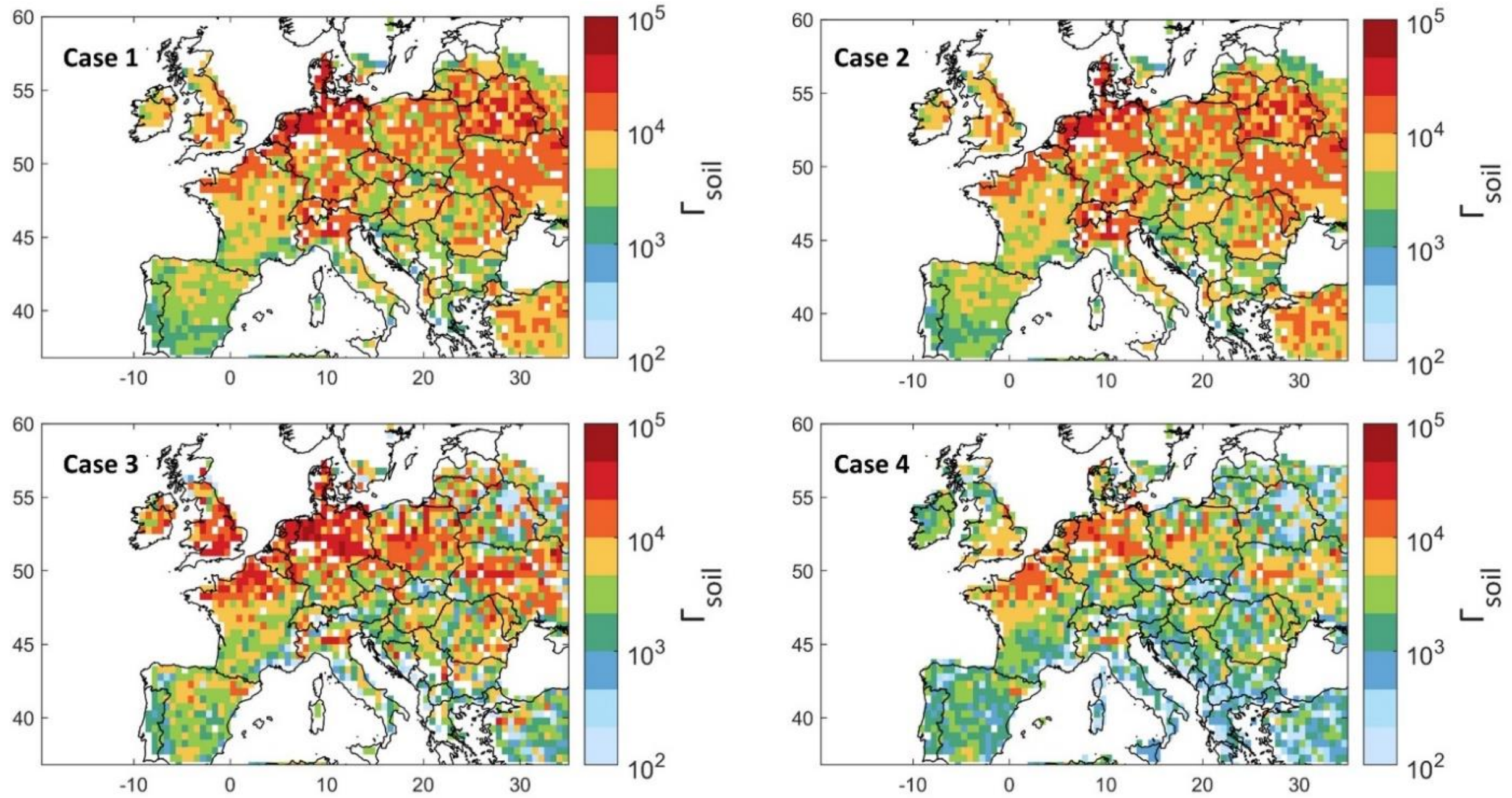

**Figure 5. Ammonia soil emission potential ($\Gamma_{soil}$) on a log10 scale from model simulation, observation and reanalysis for 4 different cases (see text for details).**

We show in supplementary material Figure S2, the emission potential (similarly to what we show in Figure 5) but from a fixed and averaged $k$ value for all land types. Figure S2 shows the importance of using a variable $k$ that is adjusted to each land type is depicted in supplementary materials (Figure S2). In Figure S2, the fixed $k$ used is calculated assuming 14 days of fertilization ($k = 10^{-3}$ m s$^{-1}$), 7 days when $k$ value reduces ($k = 10^{-5}$ m s$^{-1}$), and 10 days when $k$ is low ($k = 10^{-6}$ m s$^{-1}$) resulting in average of $k = 4.5 \times 10^{-4}$ m s$^{-1}$. The difference in the emission potential between fixed and spatially variable $k$ is shown in supplementary material Figure S3, where we see that a fixed $k$ might overestimate $\Gamma_{soil}$ by 10 to $10^3$ on a log10 scale (500 – 3000 %), in agricultural areas.

Figure 5 illustrates the ammonia soil emission potential $\Gamma_{soil}$ calculated using Eq. (2-1) and $k$ values presented in Figure 4. After assigning the variable mass transfer coefficient, the remaining variables needed to calculate $\Gamma_{soil}$ in Eq. (2-1) are ammonia concentration and lifetime, as well as the skin temperature. For this reason, the emission potential $\Gamma_{soil}$ shown in Figure 5 is calculated using different configurations.

- Case 1: GEOS-Chem ammonia and lifetime with MERRA-2 T skin, i.e. simulated $\Gamma_{soil}$,
- Case 2: GEOS-Chem ammonia and lifetime and ERA5 Tskin, to check the effect of using ERA5 vs MERRA-2 for skin temperature,
- Case 3: IASI ammonia, ERA5 T skin and GEOS-Chem ammonia lifetime,
- Case 4: IASI ammonia, ERA5 T skin and ammonia lifetime from Evangeliou et al. (2021), that were calculated using LMDz-OR-INCA chemistry transport model. The latter couples three models: The general circulation model GCM (LMDz) (Hourdin et al., 2006), the INteraction with Chemistry and

Aerosols (INCA) (Folberth et al., 2006), and the land surface dynamical vegetation model (ORCHIDEE) (Krinner et al., 2005).

Based upon the four cases, we calculate a range of emission potentials. When calculating $\Gamma_{soil}$, we filtered data points with ammonia total column concentration less than 5 × $10^{14}$ molecules $cm^{-2}$. The latter are mostly grid boxes concentrated above 56° North that we consider as noise (shown in white pixels on Figure 5).

T skin from ERA5 and MERRA-2 agree very well, with a coefficient of determination $r^2$ = 0.97 (Figure S4 in the supplementary material). This explains the excellent spatial correlation between cases 1 and 2. Since IASI-$NH_3$ retrievals use ERA5 T skin, this also suggests that using MERRA-2 or ERA5 does not affect our $\Gamma_{soil}$ calculation. In case 3, the emission potential agrees spatially and in value with that of GEOS-Chem. However, we observe higher $\Gamma_{soil}$ in regions such as Ireland, England, North France, Northeastern Spain, and Poland. This is due to the underestimation/overestimation of ammonia from GEOS-Chem as compared to IASI observations (Figure 1a). For instance, $\Gamma_{soil}$ from IASI and ERA5 (case 3) differs with that from GEOS-chem and ERA5 (case 2) by up to -70 % in the Po Valley (Italy) and +60 % in England. Looking at Table 1, this difference can be explained by the corresponding MRD for each of the regions, in which it is -64 % for England and +110 % for the Po Valley. Similarly, the differences between case 3 and 4 reach up to +66 % in England, and this is mostly due to the 10-hours difference between ammonia lifetime from GEOS-Chem and Evangeliou et al. (2021) (Figure S1 in the supplementary material). The lowest $\Gamma_{soil}$ were obtained in case 4, due to the higher lifetimes than those calculated from GEOS-Chem (Figure S1); note that $\Gamma_{soil}$ is inversely proportional to ammonia lifetime (Eq. (2-1). In fact, the longer ammonia stays in the atmosphere (longer lifetime), the less the flux will be directed from the soil to the atmosphere (less ammonia emission).

In the four cases presented in Figure 5, we see similar spatial distribution of ammonia emission potential ranging from 12 × $10^{-1}$ in a forest to 9.5 × $10^4$ in a cropland (monthly average considering all the cases). In agricultural lands, our results show that $\Gamma_{soil}$ ranges from 2 × $10^3$ to 9.5 × $10^4$. Our values for croplands start at around $10^3$. In fact, most of the studies summarized in Zhang et al. (2010) reported $\Gamma_{soil}$ that range mostly from $10^3$ to $10^4$ in fertilized croplands/grasslands; the minimum $\Gamma_{soil}$ reported is in the order of $10^2$ and the maximum is of the order of $10^5$. Therefore, our values fit within the range of $\Gamma_{soil}$ calculated in the literature and summarized in Zhang et al. (2010) and the references within. Personne et al. (2015) focused on Grignon, an agricultural region near Paris, France (48◦51′N, 1◦58′E). They obtained $\Gamma_{soil}$ values between 1.1 × $10^4$ to 5.8 × $10^6$. In the present study, the emission potential over this region is between 5 × $10^3$ (case 4) to 2 × $10^4$ (case 2). In this study, it is expected to obtain lower values than the ones measured over specific field. Therefore, we consider our results to be in good agreement with the obtained values in Personne et al. (2015), since ours reflect an average of a coarse patch of land of the size 55 × 70 $km^2$ approximately, with a 31-day mean.

The mean emission potentials per ammonia source region in Europe (shown in rectangles in Figure 2 and Figure 3) and per case are shown in Figure 6, and listed in Table 1. Table 1 shows the average lifetime from GEOS-Chem (hours), the average T skin from the three datasets that we used (°C), the average ammonia emission potential in all the cases examined (dimensionless), and the average ammonia columns from IASI and GEOS-Chem (molecules $cm^{-2}$). The four cases show a similar pattern with the North European Plain exhibiting the highest emission potential. This has been shown in Figure 1, Figure 2, and Figure 5, as well as in Table 1, where $\Gamma_{soil}$ is higher in regions with high ammonia columns. This is expected in fertilized lands (croplands), since $\Gamma_{soil}$ is proportional to the concertation of ammonia near the surface. The latter increases when the soil content in ammonium ($NH_4^+$) increases following the application of nitrogen-based fertilizers.

Figure 6 also shows that for cases 1 and 2 (GEOS-Chem) the emission potential in the Po Valley is higher as compared to case 3 (IASI), although it stays within the margin of error. This is due to the effect of temperature. Table 1 shows that at the time of the IASI overpass, T skin from ERA5 in the Po Valley is almost twice as large (8.95 °C) as the monthly averaged temperature (4.46 °C). The effect of skin temperature through Eq. (2-1) makes the emission potential highly dependent. In fact, $\Gamma_{soil}$ is both directly and inversely proportional to T skin, however, the exponential in the denominator has ~10 times more effect on the value of $\Gamma_{soil}$ than the T skin in the numerator. Therefore, through Eq. (2-1), we conclude that an increase in temperature by 1°C will reduce $\Gamma_{soil}$ by around $-8\%$.

The standard deviation (shaded area) is found to be the highest in the North European Plain, which is also the largest region (hence higher variability is expected), especially when considering case 3 with IASI. IASI distinguishes different source sub-regions, leading to higher spatial variability of ammonia, and therefore $\Gamma_{soil}$. As Figure 5 has shown, case 4 has the lowest $\Gamma_{soil}$, with a factor of two lower than cases 1 to 3. This is due to the longer lifetimes calculated by Evangeliou et al. (2021). However, we note that all the regions exhibit the same inter-variability between each of the case, regardless of the lifetimes used.

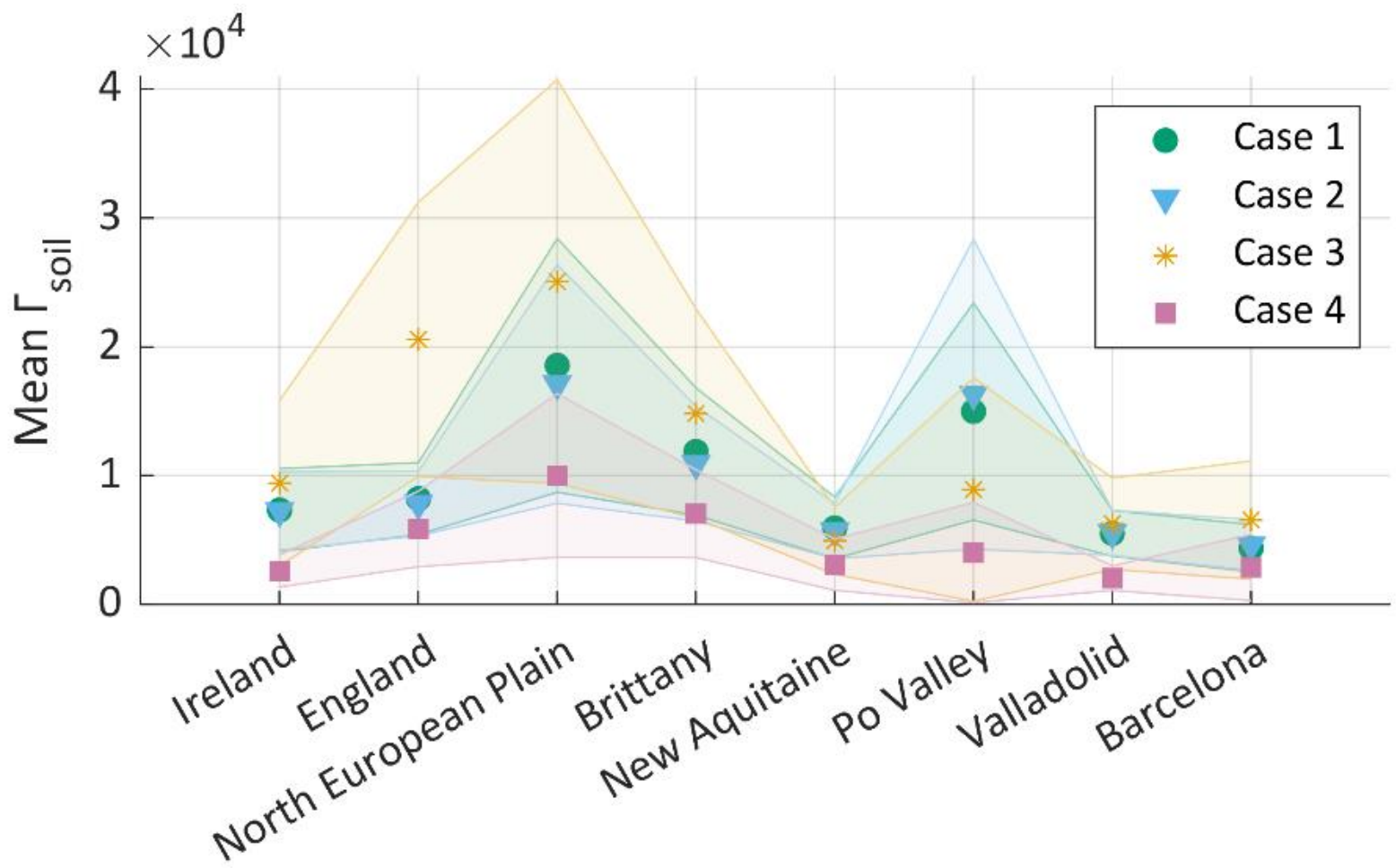

**Figure 6. Mean ammonia emission potential $\Gamma_{soil}$ per region and per case, with the error margin on the mean as the shaded area (95th percentile) for cases 1 to 4. The cases are explained in Figure 5 and its discussion.**

**Table 1.** Summary of $NH_3$ average lifetime, emission potential, concentrations and the T skin in selected regions in Europe.

| Region | Country | $\tau_{NH_3}$ [hours] | T skin [°C] | | | $\Gamma_{soil} \times 10^4$ [dimensionless] | | | | $NH_3$ concentrations [molecules × $10^{15}$ $cm^{-2}$] | | |
|---|---|---|---|---|---|---|---|---|---|---|---|---|
| | | | ERA5 IASI Overpass | ERA5 | MERRA-2 | Case 1 | Case 2 | Case 3 | Case 4 | IASI | GEOS-Chem | Mean MRD [%] |
| Ireland | Ireland | 3.34 | 8.74 | 5.78 | 6.23 | 0.73 | 0.72 | 0.94 | 0.26 | 2.5 | 1.5 | − 46 |
| England | England | 3.15 | 8.54 | 5.87 | 5.73 | 0.82 | 0.78 | 2.06 | 0.58 | 4.8 | 1.2 | − 64 |
| North European Plains | Belgium, Netherlands | 5.16 | 7.46 | 4.93 | 4.57 | 1.86 | 1.71 | 2.51 | 1.00 | 7.7 | 3.9 | − 49 |
| Brittany | France | 6.93 | 10.48 | 8.13 | 8.16 | 1.19 | 1.09 | 1.48 | 0.70 | 5.8 | 3.7 | − 60 |
| New Aquitaine | France | 8.05 | 11.25 | 7.72 | 7.47 | 0.59 | 0.57 | 0.49 | 0.30 | 4.0 | 2.9 | − 20 |
| Po Valley | Italy | 7.10 | 8.95 | 4.46 | 5.46 | 1.50 | 1.63 | 0.89 | 0.40 | 3.8 | 4.0 | + 110 |
| Valladolid | Spain | 4.53 | 11.64 | 6.87 | 6.93 | 0.55 | 0.55 | 0.62 | 0.20 | 2.5 | 1.3 | − 42 |
| Barcelona | Spain | 4.94 | 12.61 | 7.05 | 9.44 | 0.43 | 0.46 | 0.65 | 0.28 | 3.2 | 1.5 | − 49 |

464 **5. Ammonia under future scenarios**

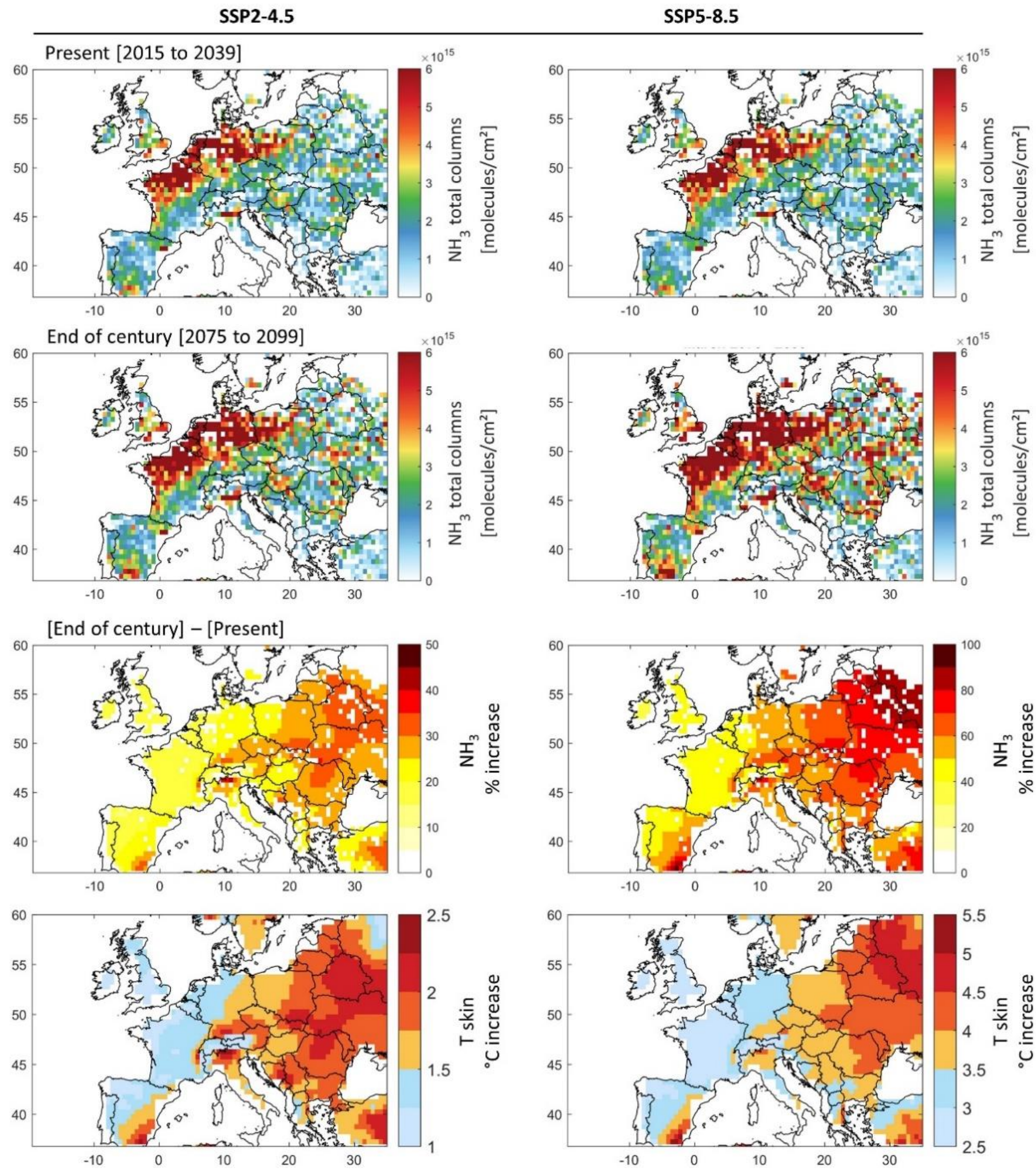

**Figure 7. First and second rows: Ammonia total column concentrations during March (monthly averages) under the present climate [2015 to 2039] (first row), and in the end of century climate [2075 to 2099] (second row), under the socio-economic scenarios SSP2-4.5 (left) and SSP5-8.5 (right). Third and fourth rows: The percentage increase in ammonia concentration (third row), and the change in T skin in °C (fourth row) by the end of the century [2075 to 2099] with respect to present climate [2015 to 2039] under SSP2-4.5 (left) and SSP5-8.5 (right). Ammonia columns were calculated using ammonia emission potential $\Gamma_{soil}$ derived from IASI and ERA5 for March 2011 (case 3), and EC-Earth T skin simulations for SSP2-4.5 and SSP5-8.5 extending from 2015 till 2099.**

As seen in Eq. (2-1), higher skin temperatures favour volatilization of ammonia from the soil. In an attempt to understand how our simplified emission potential model behaves under changing climate, as well as under future scenarios, we adopt the future T skin simulations from EC-Earth climate model, into Eq. (2-1). The two climate socio-economic scenarios that we consider are SSP2-4.5 ("middle of the road" scenario where trends broadly follow their historical patterns), and SSP5-8.5 (a world of rapid and unconstrained growth in economic output and energy use) (Riahi et al., 2017). The same Figure constructed using $\Gamma_{soil}$ from GEOS-Chem (case 1) is shown in the supplementary material as Figure S5.
We calculate current and future ammonia columns assuming that the emission potential $\Gamma_{soil}$ remains unchanged. In other words, we assume that the same amount of fertilizers and manure is used until 2100 in the agricultural fields and farms (unchanged ammonium soil content).
Figure 7 shows ammonia columns during the 25-year [2015 – 2039] representing the present climate (upper panels), and the end of the century [2075 – 2099] (middle panels). The ammonia columns in the 25-year average climate of the end of century with respect to present day climate (lower panels) are also shown.

Spatially, the present climate ammonia columns calculated from the T skin of the climate model and our emission potential from IASI (case 3 in Figure 5), agree very well with those shown in Figure 1. We do not aim at validating or directly comparing the two, as we are only interested in the climate response on ammonia concentration, i.e. by the difference due to skin temperature increase (lower panels).

From Figure 7 (lower panels) it can be seen that the increase in ammonia columns by the end of the century is more severe on the east side of Europe. Under the most likely scenario (SSP2-4.5), ammonia columns vary between +15 % in France, to around +20 % in the North European Plain (Figure 7). The largest increase is detected in Eastern Europe, where ammonia columns show an increase of up to a +50 % (Figure 7, lower left panels), creating new potential hotspots/sources of ammonia in Belarus, Ukraine, Hungary, Moldova, parts of Romania and Switzerland. Under the SSP5-8.5 scenario, the results show an increase in ammonia columns of up to +100 % in Eastern Europe (Figure 7, right lower panel). This is directly related to the higher projected increase in skin temperature over these regions. Other studies have equally reported Eastern Europe to be more affected by climate change under future scenarios, as compared to western Europe (European Environment Energy, 2022; Jacob et al., 2018). Spatially, the increase in ammonia coincides with the increase in T skin.
Figure 8 depicts the change in ammonia columns under the SSP2-4.5 and SSP5-8.5 scenarios, for our source regions (shown as rectangles in Figure 2). Ammonia columns increase is foreseen to be the highest in the Po Valley (Italy) with +26 % and +59 % under SSP2-4.5 and SSP5-8.5 respectively. It is then followed by the agricultural

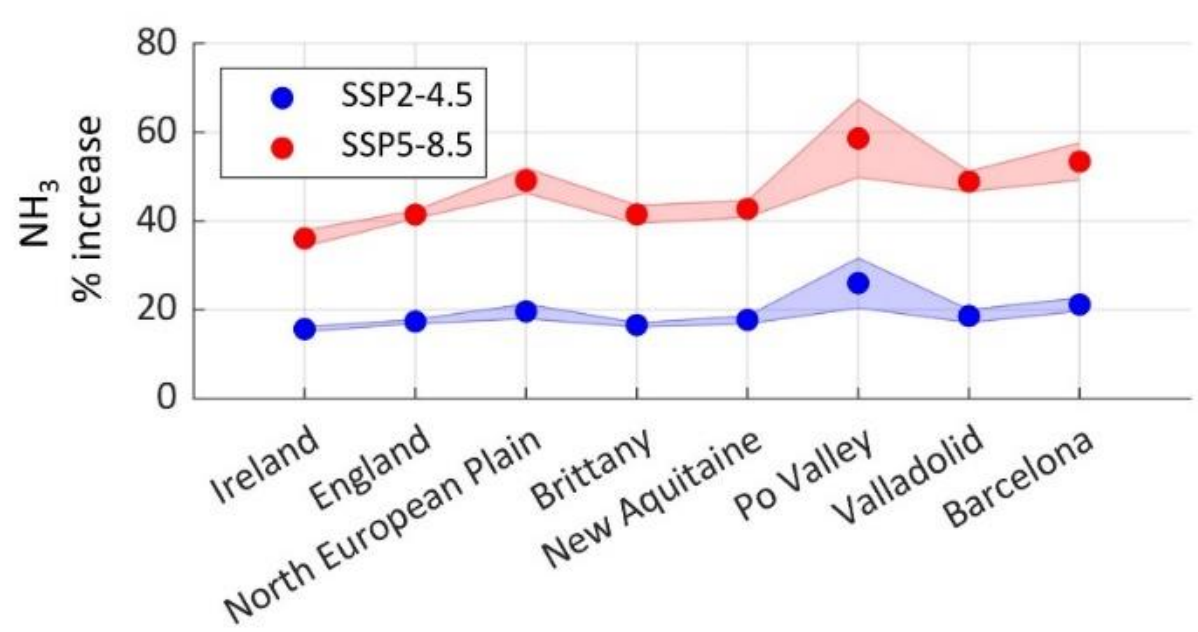

**Figure 8. The percentage increase in ammonia concentration by the end of the century [2075 to 2099] with respect to the present climate [2015 to 2039] under the two climate scenarios SSP2-4.5 (blue) and SSP5-8.5 (red), in the source regions investigated in this study. The shades around each line represent the standard deviation from the mean.**

areas around Barcelona (Spain), and the North European Plain (Belgium, Netherlands) with an increase of +21 % (+49 %) and +20 % (+53 %) respectively, under the SSP2-4.5 (SSP5-8.5) scenario. Under the SSP5-8.5, the increase in ammonia columns in percentage is more than twice the change under SSP2-4.5 (+127 % in the case of the Po Valley for instance). The Po Valley is adjacent to the Alps mountains, and due to global warming, this region is expected to experience increased evapotranspiration (Donnelly et al., 2017), which is a major factor that leads to the volatilization of ammonia.
The local and regional effect of volatilization of ammonia under different climate scenarios remains difficult to be properly assessed. Even under the "middle of the road" scenario 2-4.5, and without climate extremes (e.g. heatwaves), Europe might be facing big challenges in air or downwind agricultural regions, since chemistry and atmospheric transport (Figure 3) drive the loss of ammonia during the growing season in this part of the world.

An increase in ammonia concentration poses a significant and yet poorly understood effect on local and regional air quality through the increase in $PM_{2.5}$ concentration. We note, however, that ammonia columns in the soil are governed by a threshold. Higher temperatures will increase the rate of volatilization of ammonia from the soil, but only up to a certain point where no dissolved ammonium is left. Plants, however, can also be a source of ammonia when exposed to stressful conditions. For example, under heat stress and in instances where there are no ammonia in the air, increase in air temperature results in exponential increase in ammonia emission from plants' leaves (Husted and Schjoerring, 1996).

## 6. Discussion and conclusions

Agriculture worldwide has fed the human race for thousands of years, and will continue to do so, as mankind highly relies on it. Emissions from agricultural activities will inevitably increase, in order to meet the expected yield. In this study, we use a variety of state-of-the-art datasets (satellite, reanalysis and model simulation) to calculate the first regional map of ammonia emission potential during the start of the growing season in Europe. The emission potential can be used as a proxy to calculate ammonia columns in the atmosphere, and as such to assess its deposition, atmospheric transport, and contribution to PM formation. First, we show that the GEOS-Chem chemistry transport model is able to reproduce key spatio-temporal patterns of ammonia levels over Europe. The ammonia budget is governed by the emissions over source regions (North European Plain, Brittany and the Po valley), as well as by key loss processes. We find that chemical loss pathway is responsible of 50 % or more of the total ammonia loss over Europe. From the GEOS-Chem simulation, we calculate the average ammonia lifetime in the atmosphere which ranges between 4 and 12 hours in agricultural source regions of Europe. From this, and using the mass transfer coefficient for different land cover types, we calculate a range of emission potentials $\Gamma_{soil}$ from IASI and GEOS-Chem. We find that $\Gamma_{soil}$ ranges between from $2 \times 10^3$ to $9.5 \times 10^4$ in fertilized lands (croplands). Choosing a variable $k$ from the literature, and based on different land cover types from MODIS, we calculate $\Gamma_{soil}$ values that are consistent with those found in the literature. The increase in T skin is expected to have an effect on the emission of ammonia from the soil. Using T skin from the EC-Earth climate model, we estimate ammonia columns by the end of the century [2075 – 2099], and compare it to columns of the present climate [2015 – 2039]. Our results show that ammonia columns will double under the SSP5-8.5 scenario, and will increase by up to 50 % under the most likely SSP2-4.5 scenario. The eastern part of Europe is the most affected by the change in temperatures, and it is where we find the highest ammonia columns increase. Among the regions of focus, Italy, Spain, Belgium and the Netherlands are the most affected, as compared to France, England and Ireland. The highest increase in ammonia columns is observed in the Po Valley in Italy (+59 % under the SSP5-8.5).

We calculate ammonia concentration under future climate and during the start of the growing season (March) in Europe. However, in order to grasp the yearly budget of ammonia, it is crucial to apply this method to all seasons of the year; especially in regions with extensive agricultural activities, such as the United States, India, and China. In addition to this, more field measurements of ammonia emission potential ($\Gamma_{soil}$) in different land use / cover types are required, this can help us perform better comparison with emission potentials calculated from model and satellite data. Finally, having ammonia columns at different times of the day, from field observations or satellite measurements will allow quantification of daily emission potentials, that will in turn help us understand its diurnal variability. This will be ensured with the launch of the Infrared Sounder (IRS) on the Meteosat Third Generation (MTG) geostationary satellites scheduled in 2025.

# A. Appendix A

## 1. Ammonia-Ammonium equilibrium

Ammonia ($NH_3$) is a water-soluble gas, it undergoes protonation with $H^+$ from the hydronium ion $H_3O^+$ in an aqueous solution in order to give ammonium ($NH_4^+$ cation), the dissociation equation is expressed as follows:

$$NH_{3\,(aq)} + H_3O^+ \xleftrightarrow{K_{NH4+}} NH^+_{4\,(aq)} + HO^- \quad \textbf{(A-1)}$$

Or

$$NH_{3\,(aq)} + H^+ \xleftrightarrow{K_{NH4+}} NH^+_{4\,(aq)} \quad \textbf{(A-2)}$$

With $K_{NH4+}$ as the ammonium-ammonia dissociation equilibrium constant that can be expressed as:

$$K_{NH_4^+} = \frac{[NH_{3\,(aq)}][H^+]}{\left[NH^+_{4\,(aq)}\right]} \quad \textbf{(A-3)}$$

The solubility of ammonia in water is affected by the temperature and the acidity (pH) of the solvent (water). The equilibrium constant can be expressed as follows:

$$K_{NH_4+} = 5.67\,.\,10^{-10}\,\exp\left[-6286\,(\frac{1}{T} - \frac{1}{298.15})\right] \quad \textbf{(A-4)}$$

## 2. Henry's equilibrium

Upon its dissolution in water, $NH_3$ obeys the Henry's law. Ammonia gas ($NH_{3\,(g)}$) near the surface of the solvent is in equilibrium with the dissolved ammonia in the aqueous phase $NH_{3\,(aq)}$ (in water). Henry's equilibrium is expressed as follows:

$$NH_{3\,(g)} \xleftrightarrow{H_{NH3}} NH_{3\,(aq)} \quad \textbf{(A-5)}$$

With $H_{NH3}$ as the Henry's constant, it can be expressed as follows (Wichink Kruit, 2010):

$$H_{NH3} = \frac{[NH_{3\,(aq)}]}{[NH_{3\,(g)}]} = 5.527\,.\,10^{-4}\,.\,\exp\left[4092\,(\frac{1}{T} - \frac{1}{298.15})\right] \quad \textbf{(A-6)}$$

The partial pressure of ammonia near the surface of the soil can be calculated using Henry's constant and the dissociation equilibrium (Wichink Kruit, 2010):

$$P_{NH3} = \frac{K_{NH_4^+}\,[NH_4^+]}{H_{NH_3}\,[H^+]} = \frac{5.67\,.10^{-10}.\,\exp\left[-6286\,(\frac{1}{T}-\frac{1}{298.15})\right]}{5.527\,.\,10^{-4}\,.\,\exp\left[4092\,(\frac{1}{T}-\frac{1}{298.15})\right]} \times \frac{[NH_4^+]}{[H^+]} \quad \textbf{(A-7)}$$

If we use the ideal gas law (PV=nRT), we can draw the link between the mass density of ammonia ($NH_{3\,(g)}$) and the partial pressure:

$$\chi_{NH_3} = \frac{P_{NH_3} . M_{NH_3}}{R . T} \quad \textbf{(A-8)}$$

Where $\chi_{NH_3}$ is the concentration of $NH_3$ at the soil surface (kg m$^{-3}$), $P_{NH3}$ is the partial pressure of $NH_3$ near the surface (atm), $M_{NH3}$ is the molar mass of $NH_3$ (kg mol$^{-1}$), R is the gas constant (0.082 atm L mol$^{-1}$ K$^{-1}$), and T is the temperature in Kelvin.

Substituting Eq. (A-5) in (A-6) we get:

$$\chi_{NH_3} = \frac{2.75 .10^9 \ (\frac{gK}{m^3})}{T_{soil}} \exp\left[\frac{-1.04 .10^4}{T_{soil}}\right] \Gamma_{soil} \quad \textbf{(A-9)}$$

Where $\chi_{NH_3}$ is the concentration of ammonia at the soil surface at equilibrium (g m$^{-3}$), and is referred to as the compensation point, $T_{soil}$ is the temperature of the soil (Kelvin), $\Gamma_{NH_3}$ is the $NH_3$ emission potential from the soil and is a dimensionless ratio between [$NH_4^+$] and [$H^+$].

## 3. Ammonia total columns from IASI

In this study we use the total columns of ammonia from IASI (molecules m$^{-2}$) in order to calculate the emission potential $\Gamma_{soil}$, we should draw the link between these columns and this parameter. The bi-directional exchange of $NH_3$ between the surface and the atmosphere can be expressed by the flux (assuming a flux independent of time) (Roelle and Aneja, 2005; Zhang et al., 2010):

$$Flux_{NH_3} = k\left([NH_3]^{soil} - [NH_3]^{atm}\right) \quad \textbf{(A-10)}$$

Where $Flux_{NH_3}$is the bidirectional flux between the soil and the atmosphere (molecules (m² s)$^{-1}$), $k$ is the soil – atmosphere exchange velocity (m s$^{-1}$), also known as the mass transfer coefficient; $[NH_3]^{soil}$ is the concentration of $NH_{3(g)}$ in the soil, and $[NH_3]^{atm}$ is the concentration of $NH_{3(g)}$ in the atmosphere (molecules m$^{-3}$).

Assuming a first order dissociation of $NH_3$, we can express the change in the $[NH_3]^{col}$ total columns as follows:

$$\frac{d\,[NH_3]^{col}}{dt} = Flux_{NH_3} - k'[NH_3]^{col} \quad \textbf{(A-11)}$$

Where $k'$ is the rate of dissociation of first order $k' = {}^{1}/_{\tau}$ (m s$^{-1}$), with τ the lifetime of $NH_3$ in the atmosphere. Assuming steady state, and considering the $[NH_3]^{atm}$ as the $[NH_3]^{col}$, and $[NH_3]^{soil}$ as $\chi_{NH_3}$, Eq. (A-9) can be written as:

$$k\left(\frac{N_a . \chi_{NH_3}}{M_{NH_3}} - \frac{1}{c}\,[NH_3]^{col}\right) = \frac{[NH_3]^{col}}{\tau} \quad \textbf{(A-12)}$$

Where $c$ is the column height and is equal to 6 km. It is important to note that we neglect the effect of transport by wind since we only look at large regions. Finally, the total column of ammonia $[NH_3]^{col}$ can be written as:

$$[NH_3]^{col} = \frac{N_a \, . \, \chi_{NH_3}}{M_{NH_3} \, . \, (c + \frac{1}{k\tau})} \quad \textbf{(A-13)}$$

The column height is not considered anymore because it is negligible compared to ${}^{1}/_{k\tau}$ , using Eq. (A-6) in (A-11) we get:

$$[NH_3]^{col} = \frac{2.75\,.10^{27}\left(\frac{gK}{cm^3}\right)}{T_{soil}} \exp\left[\frac{-1.04\,.10^4}{T_{soil}}\right] \Gamma_{NH_3}\,.\,k\tau \qquad \left(\frac{molecules}{cm^2}\right) \qquad \textbf{(A-14)}$$

Note that $2.75\,.\,10^{27} = \frac{a\,.\,N_a\,.c'}{M_{NH_3}}\left(\frac{K\ molecules}{cm^3}\right)$, where $a = 2.75\,.\,10^3\ (g\ K\ cm^{-3})$, $N_a$ Avogadro's number (6.0221409 × $10^{23}$ molecules $mol^{-1}$), $10^{-2}$ is added to convert $k$ from m $s^{-1}$ to cm $s^{-1}$, and $M_{NH_3}$the molar mass of $NH_3$ (17.031 g $mol^{-1}$). The emission potential of $NH_3$ from the soil can we written as:

$$\Gamma_{soil} = \frac{[NH_3]^{col}\,.\,T_{soil}}{\exp(\frac{-b}{T_{soil}})} \frac{M_{NH_3}}{a\,.N_a\,.10^{-2}}\,.\frac{1}{k\tau} \qquad \textbf{(A-15)}$$

Where $b = 1.04\ \times 10^4\ K$.

## Author contribution

RA contributed to the conception and design of the article, developed the code, wrote the manuscript, analysed and interpreted of the data, and approved the version for submission; CV, CC, and PFC revised the manuscript; WCP provided the GEOS-Chem simulation data, and revised the manuscript; NE provided ammonia lifetime calculation using the LMDz-OR-INCA chemistry transport model and commented on the manuscript; MVD and LC contributed to the acquisition of the IASI ammonia data (NH3-v3R-ERA5), and revised the manuscript; SS contributed to the conception and design of the article, provided the EC-Earth temperature data, and revised the manuscript, and approved the version for submission.

## Acknowledgments

The IASI mission is a joint mission of Eumetsat and the Centre National d'Etudes Spatiales (CNES, France). The authors acknowledge the Aeris data infrastructure for providing the IASI L1C and L2 data.

## Funding information

Rimal Abeed is grateful to CNES for financial support. The research in Belgium was funded by the Belgian State Federal Office for Scientific, Technical and Cultural Affairs (Prodex HIRS) and the Air Liquide Foundation (TAPIR project). This work is also partly supported by the FED-tWIN project ARENBERG ("Assessing the Reactive Nitrogen Budget and Emissions at Regional and Global Scales") funded via the Belgian Science Policy Office (BELSPO). L. Clarisse is Research Associate supported by the Belgian F.R.S.-FNRS. C. Clerbaux is grateful to CNES for scientific collaboration and financial support. N. Evangeliou was funded by Norges Forskningsråd (ROM- FORSK – Program for romforskning of the Research Council of Norway (grant no. 275407)).

## Competing interests

The authors are aware of no competing interests.

## Data accessibility statement

The IASI-$NH_3$ used in this study are retrieved from the Aeris data infrastructure (https://iasi.aeris-data.fr/nh3r-era5/). ERA5 skin temperature from 1979 to present are available for download in the following DOI: 10.24381/cds.adbb2d47. The GEOS-Chem outputs used in this study are only available upon request. EC-Earth3 model output prepared for CMIP6 ScenarioMIP are retrieved here: https://doi.org/10.22033/ESGF/CMIP6.727. The MODIS land cover data are available for download in the following link: https://doi.org/10.5067/MODIS/MCD12Q1.006.

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
