# Peer review of "A roadmap to estimating agricultural ammonia volatilization over Europe using satellite observations and simulation data"

_EGUsphere, 2022_

## Referee Comment (RC1)

**Review of "Estimating agricultural ammonia volatilization over Europe using satellite observations and simulation data**

**Summary**

This paper sets ambitious goals: to estimate current ammonia volatilization over Europe from satellite data and to predict how it will change as Europe warms. These are interesting ideas, that could potentially provide useful data to the air quality community, due the strong connection between ammonia concentrations and $PM_{2.5}$ amounts. However, given the many uncertainties in the calculation of the soil emission potential (such as the mass transfer coefficient), the work in this paper is more of a roadmap on how this might be done rather than a set of reliable estimates. Nevertheless it is still a valuable paper for the community, as it demonstrates the methods and the types of datasets that are needed to achieve its stated goal.

The paper is overall well organized, though a few sections are confusing and need rewriting. Once the suggested revisions are made, the paper should be accepted for publishing.

**Technical comments**

Section 3.1:

The authors attribute some of the difference between IASI and GEOS-CHEM to different sampling times: IASI only measures NH3 at 9:30 am, while the GEOS-Chem output was averaged over an entire month. Wouldn't it be possible to eliminate this disparity by sampling GEOS-Chem only at 9:30 am? Please explain why this was not done.

Section 3.2

The authors state that the ammonia lifetime in New Aquitaine is high due to air stagnation. Why is this region prone to stagnation in March?

I do not think the authors should state that the GEOS-Chem lifetime estimates agree with the Evangeliou results, since the former range from 1 to 13 hours and the latter from 10 to 13. Please rewrite this statement.

Can the authors explain why the loss to transport in England is lower than in Ireland, even though it is also affected by the Gulf Stream?

Section 4

The text between lines 369 and 390 is extremely confusing, in part because Figure 5 is referenced before the differences between the four plots are explained. The calculation of the fixed k is also a bit hard to follow. Finally, in the caption for figure S3 it is stated that cases 1, 2

and 3 have identical soil emission potential, which is not true, since case 3 uses IASI rather than GEOS-Chem NH3. Please reorganize and rewrite this section and make it clearer.

The comparison between the soil emission potential from IASI and GEOS-Chem for the cases of England and the Po Valley uses percentages that are not consistent with the values listed in Table 1. Please explain how they were calculated. And what are the three values of Tskin listed in Table 1?

The discussion of Figure 6 (lines 435 to 448) mentions the average temperature and states that in the Po Valley Tskin from ERA5 is twice as high as this average temperature. Why is this relevant? Don't cases 1, 2 and 3 use very similar temperatures (ERA-5 or MERRA)? There is no mention of average temperature in description of the cases. Please explain.
The statement that the inter-variability between the cases does not depend on the lifetime does not seem to be true. Maybe which inter-variability needs to be defined?

Section 5

On line 472 the author state that current and future ammonia columns are calculated assuming that the emission potential is unchanged. If the whole point of the future climate modeling exercise is to look at effect of changing temperature on the volatilization of ammonia, and the emission potential is strongly dependent on temperature, this sentence does not make sense.

Appendix

The referencing of multiple equations in the appendix was wrong. I have corrected them, but please check and make sure my changes make sense. Finally, please also confirm that the calculation of the first constant in (A-9) (2.75e9) is correct. I was unable to reproduce this value, but that could be an error at my end.

**Minor edits (**suggested changes are in **bold)**

Line 42: … **amounted** to

Line 43: …a very reactive base, **and** constitutes …

Line 47: … total ammonia gas **is** believed **…**

Line 60: … as shown in the **Appendix.**

Line 61: …of ammonia in the water in the soil is **a function of** soil acidity (pH) and temperature …

Line 62: …and **controlled by** the dissociation …

Line 63: …exists in the gas phase, and therefore Henry's law **can be used to describe** ….

Line 84: …during the 2003-2019 period …

Line 85: …(2022), **leading to increased** volatilized ammonia, (**due to increase in both** nitrogen …

Line 87/88: … Between **the years** 2008 and 2018, the … columns **is estimated to be** …
Line 107: … ammonia **to provide** regional …

Line 116: **where**

Line 120: …c' is 100

Line 123: … It is **a function of** the roughness length …

Line 125: The sentence starting with "It can be explained by …" is unclear. Are the authors stating that a resistance model is used to calculate k?

Line 177: …  in areas  **where fertilizers are applied**.
         Show the emission potential where?

Line 204 : … we use the ECMWF **European Earth Consortium climate model** …

Lines 216-219:… two scenarios: **the SSPP2-4.5, a "middle of the road" socio-economic scenario with a nominal 4.5W/m2 radiative forcing level by 2100, similar  to the RCP-4.5 scenario, and the SSP5-8.5, the upper edge of the SSP scenario spectrum with a  high fossil-fuel development use the 21st century.**

Line 238: …2011), **which** marks …

Lines 239-240: …The differences **are likely due to sampling issues:  only cloud-free data are used to retrieve ammonia and different sampling times:** IASI ….

Line 248: … Therefore, **assuming that** meteorological …

Line 259: What does [not shown here] mean?

Line 275: The lifetime of ammonia () is shown in Figure 2d.

Line 285: … air stagnation in that area

Line 287: … and (AQEG, 2012**), and  these PM2.5 particles can dissociate, releasing** ammonia

Line 295: … **considered** the loss …

Line 296**: adopted** here …

Line 333: … literature. **Note that ammonia transfer coefficients are not available for all land types.**

Line 335: … in grey in Figure 4

Line 336: … and swine manure, **therefore, this value was assigned** to croplands…

Line 343: The sentence starting with "These values " should maybe be rewritten as :
These values obtained by using MODIS land cover types  and published estimates of k values represent our best effort to  realistic mass transfer coefficients, and therefore realistic soil emission potentials.

Line 348: Are the authors extrapolating or aggregating by averaging over each GEOS-Chem grid box? Please make this clear.

Lines 357-358: Maybe rewrite as: The $k$ value assigned for forests represents the SO2 exchange in high croplands; this value may be very different for  ammonia, since NH3 can easily dissolve in the water film on leaves under conditions of high humidity.

Lines 360-363: Again I think the authors mean aggregate not extrapolate. Which several grids? Isn't the MODIS grid just being aggregated to the GEOS-Chem grid?

Line 365: Using a land type …

Line 392:  What is the sentence starting with "Based upon …" supposed to convey?

Line 399: … The emission potential does not agree in value with that of GEOS-Chem

Line 407:  … England, **northern** France, **northeastern** Spain and Poland….

Line 412: …potential **with values** ranging from ….

Line 413: Are croplands different from agricultural lands? If not, the sentence starting with "Our values" seems unnecessary.

Lines 420-423: In this study, **lower values than those measured in the field are expected. Therefore, we consider our results to be in good agreement with the values in Personne et al. (2015), since ours reflect a 31 day mean of an average of over a large area (55x70 km$^2$)** .

Line 433: …soil content **of** …

Caption of Figure 6: … are explained **in the discussion on Figure 5.**

Line 485: …more severe in **eastern** Europe ….

Line 487: …up to +50%...

Line 505: … facing big challenges **in air (??)  or downwind of large** agricultural regions ….

Line 512: …where there **is** no ammonia …

Appendix

Line 569: **where  $H_{NH3}$ is Henry's constant, which** can be …

Line 580: Substituting Eq.(A-7) into (A-8) we get:

Line 586: **Since** in …

Line 591: … **where** Flux$_{NH3}$ …

Line 597: .. **Eq. (A-11)** can be written as ..

Line 603: … using **Eq. (A-9) in (A-13)** we get:

---

## Author Comment (AC1)

We would like to thank reviewer #1 for their constructive feedback on the manuscript and useful comments, questions and suggestions, which all have been addressed. We believe the manuscript has been improved that way. Point-by-point responses are provided below. The original review comments are shown in black, our responses are shown in blue.

The article has estimated ammonia emission potentials from agricultural land using satellite remote sensing data and a chemical transport model. The analysis was done across the European continent. Future implications of climate change on emission potentials are also analysed. I am not expressing my opinion about the article acceptance at this stage. The other reviewers might have an opinion in this regard. However, after reading the article, I have following major concerns.

**Major concerns**

1. The study period seems to be only the month of March 2011 which marks the starts of growing season (Line 26-27). Since the equation (2-1) points a direct relationship between the ammonia emission potential ($\Gamma_{soil}$) and the soil temperature.

   How much monthly skin temperature variations are there across the Europe?

   Skin temperature in Europe varies with a standard deviation on the daily average that is mostly between 2 and 6°C, in Northern, Central, Western and Southwestern Europe. And between 4 to 8°C in Eastern Europe. Review Figure 1, shows the standard deviation on the daily average for the month of March 2011 in Europe, calculated from ERA5.

[Figure]

   Review Figure 1. The standard deviation on the daily average for the month of March 2011 in Europe.

   The following sentence was added in section 2.2: "Skin temperature in Europe varies with a standard deviation on the daily average that is mostly between 2 and 6°C, in Northern, Central, Western and South-western Europe. And between 4 to 8°C in Eastern Europe (not shown here)."

   **What are the fertilization practices in the region? Are there any seasonal variations of fertilizer application rates?**

[Figure]

Synthetic fertilizer application (kg N/ha/yr)

Review Figure 2. Average rates of synthetic N fertilizer applied to cropland and permanent grassland. The figure shows all countries where more than 3% of the cumulative N fertilizer use has been applied to permanent grassland. The top left panel shows results for the 22 present-day countries which this study covers 1961–2019. Figure is from Einarsson et al. (2021).

It is not easy to get accurate information on the frequency of fertilizers application in all Europe. However, a survey conducted in 2011 mentioned that the fertilizer application in France takes place one to four times per season, according to the crop type (Agreste, 2014). While we do not have information on the practices in other European countries, we show below the N-fertilizers application per surface area in the EU-28 countries. Review Figure 2 is the synthetic fertilizer application (kg N/ha/yr) in different European countries, the figure is from Einarsson et al. (2021). We can see that the N applied per surface area is quite stabilized after year 1980, with some fluctuations from year to year in most countries. To answer the question about the seasonal variation of fertilizer application, yes, the

application can change from year to year. But the fluctuations are less pronounced between year 2000 and 2020 as the graph shows (Review Figure 2).

Ammonia concentrations peak twice a year, during summer and spring. Van Damme et al. (2022) studied the weekly seasonal variation of ammonia concentrations in Europe, and found that ammonia peaks during the weekdays preceding the weekend. For instance, in the Po Valley in Italy, ammonia peaks on Saturday and starts decreasing on Sunday. In the Ebro Valley in Northwestern Europe, ammonia peaks on Thursday-Friday, and the decrease starts again on Sunday (Van Damme et al., 2022).

These aspects are missing in the article. If you consider all these aspects to its minimum, the logic behind one month simulation and drawing future changes (%) in $\Gamma_{soil}$ is not justified.

The future estimation of ammonia considers only March under different socioeconomic scenarios (SSPs), although March doesn't reflect the whole spring season (March-April-May), it can however tell us how the season will start.

Ammonia volatilization from the soil is enhanced by higher temperatures, and the fluctuations of other meteorological parameters. We use simulation data for the month of March during 2011, because during that period Europe witnessed isolated and connected ammonia and particulate matter pollution episodes that were in part due to intensive fertilization during this month (Viatte et al., 2022). The second reason is that the fertilizers spreading activities start in March in Europe, as shown by the FAO NDVI (e.g. start of the growing season in Ireland (FAO Earth Observation, 2022)). Ammonia concentrations are expected to increase during March every year, due the increase in air and land temperatures and decrease in precipitations (as compared to February). Since wet deposition is considered a sink of atmospheric $NH_3$.

Moreover, we now changed the title of the last section "The effect of temperature change on the volatilization of ammonia" in order to correctly reflect its content: it is meant to show a case study of what would be $NH_3$ concentrations given *only* the increase in temperature due to climate change.

What type of fertilizers are used in the region? How much $NH_3$ content each fertilizer has?

The $NH_3$ is a by-product of the fertilizer application, and its concentration depends, as we show here, on many factors, in the soil and in the near surface.

However, we note that in Europe, roughly 90 % of the mineral fertilizers used are nitrogen-based (N-fertilizers), and 10 % are phosphorus-based (Review Figure 3). Among the N-fertilizers used, urea (22 %) and nitrate fertilizers (45 %) dominate the market in the 27 EU countries (Fertilizers Europe, 2016). These two will release $NH_3$.

[Figure]

Review Figure 3. Estimated mineral fertiliser consumption by agriculture in the EU-27, 2008-2018 (European Environmental Agency, 2022). Figure and data can be accessed via the following link: https://www.eea.europa.eu/data-and-maps/figures/estimated-mineral-fertiliser-consumption-by.

Moreover, in the context of the study, we are not concerned about the fertilizer content since we derive the emission potential values directly from the atmospheric concentration of $NH_3$. The method we use does not require the information about the fertilizer content.

**To address the previous comment, the following paragraphs were added to the manuscript:**

"Around 90 % of the mineral fertilizers used in Europe are nitrogen-based, with urea and nitrate fertilizers dominating the market in the 27 EU countries, since they make up 22 % and 45 % of the total market (Fertilizers Europe, 2016)."

"The frequency of fertilizers application can vary per crop type and per country, as well as from year to year. In Europe, however, the N applied per surface area is quite stabilized after year 1980, with some interannual fluctuations in most European countries (Einarsson et al., 2021). As to our knowledge, accurate information on the application frequency per country is not reported. While the application frequency can change from year to year, the fluctuations are less pronounced after the year 2000. For instance, in France and Belgium the nitrogen content fluctuates between 100 and 110 kg N/ha/year, from 2000 to 2020 (Einarsson et al., 2021)."

Why did authors choose to assigned mass transfer coefficient ($k$) values to non-fertilized forests, shrublands and grasslands that ultimately resulted in $\Gamma_{soil}$ ranges of $10 - 10^2$ in a non-fertilized soil. The justification given between lines (337-343) require some literature-based support to establish the linkage between $SO_2$ and $NH_3$ $k$ values over non- fertilized land-use.

New explanation is added to the addressed paragraph (lines 334-343). Now it reads as the following:

"For water bodies and other land types that are not considered here (see Sect. 2.2), the mass transfer values $k$ were set to zero and represented in grey colour in Figure 4. In a laboratory experiment, Svensson et al. (1993) reported $k = 4.3 \times 10^{-3}$ m s$^{-1}$ for a mixture of soil and swine manure, as therefore, we assign this value to croplands. Due to the lack of NH$_3$ $k$ values for non-fertilized forests, shrublands and grasslands in the literature, we used values originally assigned for SO$_2$, bearing in mind that these are approximate values and they reflect mostly the conditions of the soil cover type (short, medium or tall grass) rather than the gas itself. In Aneja et al. (1986), the authors estimated the mass transfer coefficient for both NH$_3$ and SO$_2$ above different types of crops, they found similar values. For NH$_3$, $k$ varied between 0.3 and 1.3 cm s$^{-1}$, and for SO$_2$ it varied between 0.5 and 1.5 cm s$^{-1}$ (Aneja et al., 1986). Since the latter study estimates several values for NH$_3$ mass transfer coefficient, over different types of crops, we will use the $k$ provided by Svensson et al. (1993), since it is better adapted to reflect NH$_3$ emission from fertilizers, and is not dependent on the crop type. To assign a $k$ value for forests, we used values reported in Aneja (1986) ($k = 2 \times 10^{-2}$ m s$^{-1}$), which originally represent deposition velocity (mass transfer) of SO$_2$ in a forest (high crops), since both SO$_2$ and NH$_3$ showed similar $k$ in above crops. For shrublands and grasslands (the two land types have the same $k$), we used the value $k = 8 \times 10^{-3}$ m s$^{-1}$ that has been reported in Aneja et al. (1986) as the deposition velocity (mass transfer) of SO$_2$ in a grassland (medium crops)."

2. Any bias correction of SSP scenarios was done before analysing future climate change implications on $\Gamma_{soil}$?. If so, kindly mention it in the article.

   No bias correction was done.

3. The study is based on number of assumptions e.g., assuming $[NH3]_{atm}$ equals to $[NH3]_{col}$ (line 597). The questions raised above are also based on assumptions. So, it would be better to discuss the limitations of this analysis thoroughly in a separate sub-section.

   Most of the atmospheric NH$_3$ is present near the surface in the lower boundary layer (Dammers et al., 2019), that is why we can say that $[NH3]_{atm}$ is equal to $[NH3]_{col}$. This information was added to the Appendix in order to justify this assumption: "[…]; $[NH_3]^{soil}$ is the concentration of NH$_{3(g)}$ in the soil, and $[NH_3]^{atm}$ is the concentration of NH$_{3(g)}$ in the atmosphere near the surface (molecules m$^{-3}$). We can consider that $[NH_3]^{atm}$ is identical to the total column of NH$_3$ provided by IASI and denoted here as $[NH_3]^{col}$. This is because most of the atmospheric NH$_3$ are located in the lower boundary layer (Dammers et al., 2019)."

**Minor concerns**

1. Paragraph two (lines 52-56) is too short either expand it or merge it with adjacent paragraphs.

   We merged paragraph two with the previous paragraph. It now reads as "Soils are known to be a source of atmospheric ammonia, especially in areas of intensive agricultural

practices (Schlesinger and Hartley, 1992), and this is due to enriching the soil with the reactive nitrogen present in fertilizers. The increase in the application of synthetic fertilizers, and intensification of agricultural practices is believed to be the dominant factor of the global increase in ammonia emissions over the past century (Behera et al., 2013; McDuffie et al., 2020). In addition to agriculture, ammonia can be emitted from industrial processes, biomass burning (Van Damme et al., 2018), and natural activities such as from seal colonies (Theobald et al., 2006)."

1. There is no Result section it would be better to term section 3 as Results rather than "GEOS-Chem model simulation: validation and analysis".

   The title of section 3 was changed to "Results and discussions".

2. Discussion and conclusion should be separate.

   The title "discussion and conclusions" was replaced by "conclusions", since the last section reflects the conclusion and the results cited listed, rather than the discussion.

3. The sentence structure and grammatical errors can be rectified by employing some professional services. I would highly recommend that.

   we made changes throughout the manuscript, we hope that this rectifies the sentence structure.

**References**

Agreste. (2014). *Enquête Pratiques culturales 2011—Grandes cultures et prairies | Agreste, la statistique agricole* (No. 21; Agreste Les Dossiers). Ministère de l'Agriculture, de l'Agroalimentaire et de la Forêt. https://agreste.agriculture.gouv.fr/agreste-web/disaron/dos21/detail/

Aneja, V. P., Rogers, H. H., & Stahel, E. P. (1986). Dry Deposition of Ammonia at Environmental Concentrations on Selected Plant Species. *Journal of the Air Pollution Control Association*, *36*(12), 1338–1341. https://doi.org/10.1080/00022470.1986.10466183

Dammers, E., McLinden, C. A., Griffin, D., Shephard, M. W., Van Der Graaf, S., Lutsch, E., Schaap, M., Gainairu-Matz, Y., Fioletov, V., Van Damme, M., Whitburn, S., Clarisse, L., Cady-Pereira, K., Clerbaux, C., Coheur, P. F., & Erisman, J. W. (2019). $NH_3$ emissions from large point sources derived from CrIS and IASI satellite observations. *Atmospheric Chemistry and Physics*, *19*(19), 12261–12293. https://doi.org/10.5194/acp-19-12261-2019

Einarsson, R., Sanz-Cobena, A., Aguilera, E., Billen, G., Garnier, J., van Grinsven, H. J. M., & Lassaletta, L. (2021). Crop production and nitrogen use in European cropland and grassland 1961–2019. *Scientific Data*, *8*(1), Article 1. https://doi.org/10.1038/s41597-021-01061-z

FAO Earth Observation. (2022). *Crop-growing season in Ireland*. https://www.fao.org/giews/earthobservation/country/index.jsp?lang=en&code=IRL#

Fertilizers Europe. (2016). *Infinite Fertilizers—Nutrient Stewardship*.
https://www.fertilizerseurope.com/wp-
content/uploads/2019/08/Nutrient_Stewardship_Sept_2016_websize.pdf

Svensson, L., & Ferm, M. (1993). Mass Transfer Coefficient and Equilibrium Concentration as Key
Factors in a New Approach to Estimate Ammonia Emission from Livestock Manure.
*Journal of Agricultural Engineering Research*, *56*(1), 1–11.
https://doi.org/10.1006/jaer.1993.1056

Van Damme, M., Clarisse, L., Stavrakou, T., Wichink Kruit, R., Sellekaerts, L., Viatte, C., Clerbaux,
C., & Coheur, P.-F. (2022). On the weekly cycle of atmospheric ammonia over European
agricultural hotspots. *Scientific Reports*, *12*(1), Article 1. https://doi.org/10.1038/s41598-
022-15836-w

Viatte, C., Abeed, R., Yamanouchi, S., Porter, W. C., Safieddine, S., Van Damme, M., Clarisse, L.,
Herrera, B., Grutter, M., Coheur, P.-F., Strong, K., & Clerbaux, C. (2022). $NH_3$
spatiotemporal variability over Paris, Mexico City, and Toronto, and its link to $PM_{2.5}$
during pollution events. *Atmospheric Chemistry and Physics*, *22*(19), 12907–12922.
https://doi.org/10.5194/acp-22-12907-2022

---

## Author Comment (AC2)

We would like to thank reviewer #2 for their constructive feedback on the manuscript and useful comments, questions and suggestions, which all have been addressed. We believe the manuscript has been improved that way. Point-by-point responses are provided below. The original review comments are shown in black, our responses are shown in blue.

Review of "Estimating agricultural ammonia volatilization over Europe using satellite observations and simulation data

**Summary**

This paper sets ambitious goals: to estimate current ammonia volatilization over Europe from satellite data and to predict how it will change as Europe warms. These are interesting ideas, that could potentially provide useful data to the air quality community, due the strong connection between ammonia concentrations and PM2.5 amounts. However, given the many uncertainties in the calculation of the soil emission potential (such as the mass transfer coefficient), the work in this paper is more of a roadmap on how this might be done rather than a set of reliable estimates. Nevertheless, it is still a valuable paper for the community, as it demonstrates the methods and the types of datasets that are needed to achieve its stated goal.

The paper is overall well organized, though a few sections are confusing and need rewriting. Once the suggested revisions are made, the paper should be accepted for publishing.

**Technical comments**

Section 3.1:
The authors attribute some of the difference between IASI and GEOS-Chem to different sampling times: IASI only measures NH3 at 9:30 am, while the GEOS-Chem output was averaged over an entire month. Wouldn't it be possible to eliminate this disparity by sampling GEOS-Chem only at 9:30 am? Please explain why this was not done.

For the comparison of IASI and GEOS-Chem data, we rerun the same simulation with hourly output and used the GEOS-Chem output between 8:30 and 11:30 UTC, which corresponds to IASI's crossing time above Europe (and what was done in other studies e.g. Viatte et al., 2022). Therefore, the GEOS-Chem data in the latest version of the paper, are spatially and temporally coincident with those IASI measurements. We added the following text (shown below in bold) in the first paragraph of section 3.1 in the manuscript:

"**We compare those to the IASI total columns of ammonia gridded on the same horizontal resolution (0.5° × 0.625°) and over the same month. We applied spatial and temporal coincidence criteria to GEOS-Chem outputs in order to compare them with IASI morning**

**observations. For this, we selected data between 8:30 and 11:30 UTC in the GEOS-Chem model output and only considered grid cells where IASI have observations**."

Review Figure 1 shows the difference between $NH_3$ total columns from IASI and GEOS-Chem, during March 2011 in the regions of study. We compare $NH_3$ total columns from GEOS-Chem where only morning outputs are considered, those range from 8:30 until 11:30 UTC (GC - Morning), and where all hours of the day are averaged to calculate a monthly average of March 2011 (GC - Day), and finally IASI morning measurements are equally shown. If we compare $NH_3$ from GC – Morning to $NH_3$ from GC – Day, we can see that the difference is negligible. Most of the $NH_3$ is generated during the morning hours, and not throughout the whole day, therefore, using GC – Morning or GC – Day, should not make a remarkable difference in the analysis of the results. But, GC – Morning was used since it makes more sense to compare it to IASI morning $NH_3$ measurements. As such all of the related figures in the manuscript are now updated.

[Figure]

Review Figure 1. $NH_3$ total columns from IASI and GEOS-Chem, during March 2011 in the regions of study. Blue: IASI $NH_3$ total columns, red: GEOS-Chem (GC) morning average (8:30 to 11:30 UTC), and black: GEOS-Chem daily average (all hours of the day), with the corresponding standard deviation of each as lines.

Section 3.2:
The authors state that the ammonia lifetime in New Aquitaine is high due to air stagnation. Why is this region prone to stagnation in March?

New Aquitaine is located in Central Europe, this is where the highest number of stagnant days is observed during the spring season (Garrido-Perez et al., 2018). We added this information to the manuscript, now the sentence reads as: "The latter can be related to the high

probability of air stagnation is in that area, especially during spring, in comparison to Northern Europe (Garrido-Perez et al., 2018) […].”

I do not think the authors should state that the GEOS-Chem lifetime estimates agree with the Evangeliou results, since the former range from 1 to 13 hours and the latter from 10 to 13. Please rewrite this statement.

We edited the sentence, now it reads as: “Our results agree with the literature suggesting a residence time between a few hours to a few days (Behera et al., 2013; Pinder et al., 2008). We note that Evangeliou et al. (2021) estimated the lifetime of ammonia over Europe using a different model and the results showed a monthly average ranging from 10 to 13 hours.”

Can the authors explain why the loss to transport in England is lower than in Ireland, even though it is also affected by the Gulf Stream?

Although the gulf stream affects the loss to transport in England, the chemical loss is the dominant one. Acids, such as ($HNO_3$) and ($H_2SO_4$), in the atmosphere will react with ammonia ($NH_3$) since the latter is an alkaline (basic) gas. Therefore, high atmospheric concentrations of $NO_2$ and $SO_2$ (from which $HNO_3$ and $H_2SO_4$ are derived respectively), induce higher loss of ammonia to chemical reactions. In England, the annual concentration mean of both $NO_2$ and $SO_2$ are higher than in Ireland ([https://www.eea.europa.eu/themes/air/interactive/no2](https://www.eea.europa.eu/themes/air/interactive/no2), [https://www.eea.europa.eu/themes/air/interactive/so2](https://www.eea.europa.eu/themes/air/interactive/so2)). This can explain why the largest proportion of $NH_3$ is lost to chemistry in England, in spite of the effect of the gulf stream.

We added the following text to the manuscript: “Although the gulf stream also affects the loss to transport in England (region B), the chemical loss is the dominant one. Acids, such as $HNO_3$ and $H_2SO_4$ react with ammonia in the atmosphere. Therefore, high atmospheric concentrations of $NO_2$ and $SO_2$ (from which $HNO_3$ and $H_2SO_4$ are derived respectively), induce higher loss of ammonia to chemical reactions. In England, the annual concentration mean of both $NO_2$ and $SO_2$ are higher than in Ireland (European Environment Agency, 2017a, 2017b). This can explain why the largest proportion of $NH_3$ is lost to chemistry in England, in spite of the effect of the gulf stream.”

Section 4
The text between lines 369 and 390 is extremely confusing, in part because **Figure 5 is referenced before the differences between the four plots are explained.** The calculation of the fixed k is also a bit hard to follow.

This section was re-organized. Now it reads as the following: “Figure 5 illustrates the ammonia soil emission potential $\Gamma_{soil}$ calculated using Eq. (2-1) and $k$ values presented in Figure 4. After assigning a variable mass transfer coefficient, the remaining variables needed to calculate $\Gamma_{soil}$ in Eq. (2-1) are ammonia concentration and lifetime, as well as the skin

temperature. Therefore, the emission potential $\Gamma_{soil}$ shown in Figure 5 is calculated using different configurations:

1- Case 1: GEOS-Chem ammonia and lifetime with MERRA-2 T skin, i.e. simulated $\Gamma_{soil}$,
2- Case 2: GEOS-Chem ammonia and lifetime and ERA5 Tskin, to check the effect of using ERA5 vs MERRA-2 for skin temperature,
3- Case 3: IASI ammonia, ERA5 T skin and GEOS-Chem ammonia lifetime,
4- Case 4: IASI ammonia, ERA5 T skin and ammonia lifetime from Evangeliou et al. (2021), that were calculated using LMDz-OR-INCA chemistry transport model. The latter couples three models: The general circulation model GCM (LMDz) (Hourdin et al., 2006), the INteraction with Chemistry and Aerosols (INCA) (Folberth et al., 2006), and the land surface dynamical vegetation model (ORCHIDEE) (Krinner et al., 2005).

We show in supplementary material Figure S2, the emission potential (similarly to what we show in Figure 5) but from a fixed and averaged $k$ value for all land types. Figure S2 shows the importance of using a variable $k$ that is adjusted to each land type is depicted in supplementary materials (Figure S2). To calculated a fixed $k$ (common to all land types) we assume 14 days of fertilization ($k = 10^{-3}$ m s$^{-1}$, e.g. croplands), 7 days when $k$ value is reducing ($k = 10^{-5}$ m s$^{-1}$), and 10 days when $k$ is low ($k = 10^{-6}$ m s$^{-1}$, e.g. forests) resulting in average of $k = 4.5 \times 10^{-4}$ m s$^{-1}$. The difference in the emission potential between fixed and spatially variable $k$ is shown in supplementary material Figure S3, where we see that a fixed $k$ might overestimate $\Gamma_{soil}$ by 10 to $10^3$ on a log10 scale (500 – 3000 %), in agricultural areas."

Finally, in the caption for figure S3 it is stated that cases 1, 2 and 3 have identical soil emission potential, which is not true, since case 3 uses IASI rather than GEOS-Chem NH$_3$. Please reorganize and rewrite this section and make it clearer.

Figure S3 shows the difference between using a fixed $k$ value and a variable one. To calculate this difference, we use the following equation: $\frac{\Gamma_{fixed\,k} - \Gamma_{variable\,k}}{\Gamma_{variable\,k}} \times 100$. Using this equation, all other parameters are cancelled out (including the NH$_3$ concentrations from IASI/GEOS-Chem), except for $k$ and $\tau$. The resulting equation is $(\frac{k_v \tau}{k \tau} - 1) \times 100$. Case 1, 2, and 3 use the same $\tau$ NH$_3$ lifetime from GEOS-Chem, and case 4 uses $\tau$ from Evangeliou et al. (2021). Therefore, the left panel of Figure S3 is not an average of case 1, 2, and 3, but rather the relative difference that is identical for all three cases.

The caption of Figure S3 was changed to: "Relative difference (%) of $\Gamma_{soil}$ between using a fixed and a variable $k$ value: $\frac{\Gamma_{fixed\,k} - \Gamma_{variable\,k}}{\Gamma_{variable\,k}} \times 100$. Using this equation, all other parameters are cancelled out (including the NH$_3$ concentrations from IASI/GEOS-Chem), except for $k$ and $\tau$. The resulting equation is $(\frac{k_v \tau}{k \tau} - 1) \times 100$. Case 1, 2, and 3 use the same $\tau$

NH$_3$ lifetime from GEOS-Chem, and case 4 uses $\tau$ from Evangeliou et al. (2021). The left panel is the relative difference that is identical for cases 1, 2 and 3 since they use the same $\tau$ NH$_3$."

The comparison between the soil emission potential from **IASI and GEOS-Chem for the cases of England and the Po Valley uses percentages that are not consistent with the values listed in Table 1**. Please explain how they were calculated. And what are the three values of Tskin listed in Table 1?

The MRD shown in Table 1 is calculated in three main steps:

1- A spatial MRD$_{grid\_Europe}$ is calculated for Europe, resulting in what we see in Figure 1d. This is calculated following the equation: $MRD\ (\%) = \dfrac{(GeosChem\ NH_3 - IASI\ NH_3) \times 100}{IASI\ NH_3}$

2- To get an MRD for each of our regions (shown in rectangles in Figure 2a), we then extract the MRD based on longitude and latitude limits specific to each region of focus. The result is a grid of MRD$_{grid\_region}$ (e.g. MRD grid matrix for the Po Valley).

3- The MRD$_{grid\_region}$ is then averaged to give one value for the region of focus (e.g. Po Valley), and the result is shown in Table 1.

It is true that the values for England and the Po Valley do not reflect the difference between the concentrations of IASI and GEOS-Chem, shown equally in the Table 1. We calculated MRD in two additional different ways, and now the new values are more reasonable.

In Review Figure 2, we show on the left panel the NH$_3$ concentrations from GEOS-Chem (GC) and IASI, and on the right panel the MRD calculated in three different ways:

- In blue: "mean" MRD = the mean of MRD$_{grid\_region}$ (what is shown in Table 1 in the article).
- In red: "median" MRD = the median of MRD$_{grid\_region}$.
- For the third way (in black), we calculated the MRD using directly the averages of NH$_3$ for the whole region, these are the NH$_3$ concentration averages shown in Table 1 (e.g. $4.8 \times 10^{15}$ molecules cm$^{-1}$ for England). The equation is: "MRD$_d$" = $\dfrac{(mean\ GeosChem\ NH_3 - mean\ IASI\ NH_3) \times 100}{mean\ IASI\ NH_3}$.

[Figure]

Review Figure 2. Left: NH₃ concentrations in the regions of focus from GEOS-Chem (GC) in blue and IASI in red, during March 2011. Right: the mean relative difference MRD [%] as a mean (blue), median (red), and direct MRD_d (black, refer to the text for the details of calculation).

The standard deviation on the mean of MRD shown in blue is greater than 400 % (Review Figure 2, right), therefore it is better to use the median (red) or MRD_d (black) in this case. This huge difference is only seen in the region covering the Alps mountains, where we observe high differences of T skin from MERRA-2 and ERA5. We replaced the MRD values in Table 1 by the median to address this issue.

Table 1. Summary of ammonia average lifetime, emission potential, concentrations and the T skin in selected regions in Europe.

| Region | Country | $\tau_{NH_3}$ [hours] | T skin [°C] | | $\Gamma_{soil} \times 10^4$ [dimensionless] | | | | NH₃ concentrations [molecules × $10^{15}$ cm⁻²] | | |
| --- | --- | --- | --- | --- | --- | --- | --- | --- | --- | --- | --- |
| | | | ERA5 9:30 UTC | MERRA-2 8:00 to 10:00 UTC | Case 1 | Case 2 | Case 3 | Case 4 | IASI | GEOS-Chem | Mean MRD [%] |
| Ireland | Ireland | 3.34 | 8.74 | 6.23 | 0.72 | 0.44 | 0.94 | 0.26 | 2.5 | 1.4 | − 45 |
| England | England | 3.15 | 8.54 | 5.73 | 0.63 | 0.44 | 2.06 | 0.58 | 4.7 | 1 | − 79.2 |
| North European Plain | Belgium, Netherlands | 5.16 | 7.46 | 4.57 | 1.22 | 0.95 | 2.51 | 1.00 | 7.6 | 3.5 | − 55 |
| Brittany | France | 6.93 | 10.48 | 8.16 | 0.98 | 0.66 | 1.48 | 0.70 | 5.8 | 3.2 | − 43.2 |
| New Aquitaine | France | 8.05 | 11.25 | 7.47 | 0.46 | 0.32 | 0.49 | 0.30 | 4.0 | 2.6 | − 34.1 |
| Po Valley | Italy | 7.10 | 8.95 | 5.46 | 0.90 | 0.86 | 0.89 | 0.40 | 4.0 | 3.8 | + 0.1 |
| Valladolid | Spain | 4.53 | 11.64 | 6.93 | 0.46 | 0.25 | 0.62 | 0.20 | 2.5 | 1.1 | − 57 |
| Barcelona | Spain | 4.94 | 12.61 | 9.44 | 0.31 | 0.25 | 0.65 | 0.28 | 3.2 | 1.4 | − 57.5 |

The discussion of Figure 6 (lines 435 to 448) mentions the average temperature and states that in the Po Valley Tskin from ERA5 is twice as high as this average temperature. **Why is this relevant? Don't cases 1, 2 and 3 use very similar temperatures (ERA-5 or MERRA)?** There is no mention of average temperature in description of the cases. Please explain.

In the updated version of the manuscript we use NH₃ hourly concentrations from GEOS-Chem (8:30 to 11:30 UTC). Therefore, to calculate $\Gamma_{soil}$ from IASI NH₃ (case 3 and 4) and GEOS-Chem (case 2), we use T skin from ERA5 that coincides with the overpass of IASI. The new values for $\Gamma_{soil}$ for cases 1, 2 and 3 compare well in the Po Valley (see Table 1 in the previous answer).

We added the following sentence to highlight this result: "Figure 6 also shows that for cases 1 and 2 (GEOS-Chem) the emission potential in the Po Valley is almost equal to case 3 (IASI), with $\Gamma_{soil}$ = 0.9 and 0.86 × 10⁴ in cases 1 and 2, and 0.89 × 10⁴ in case 3 (see Table 1)."

And we added further explanation to the manuscript t: "To calculate $\Gamma_{soil}$ from IASI NH₃ (case 3 and 4), we used T skin from ERA5 that coincides with the overpass of IASI. We used the same T skin values from ERA5 for case 2, in which we use NH₃ hourly concentrations from GEOS-Chem (8:30 to 11:30 UTC). The ERA5 T skin are shown in Table 1."

The statement that the inter-variability between the cases does not depend on the lifetime does not seem to be true. Maybe which inter-variability needs to be defined?

This statement was removed.

Section 5

On line 472 the author state that current and future ammonia columns are calculated assuming that the emission potential is unchanged. If the whole point of the future climate modelling exercise is to look at effect of changing temperature on the volatilization of ammonia, and the emission potential is strongly dependent on temperature, this sentence does not make sense.

The $\Gamma_{soil}$ we calculated in this study (cases 1 to 4) is directly proportional to the soil content of NH₄⁺. Although, we did not need the concentration of NH₄⁺ in order to calculate it, the result reflects the amount of fertilizers added (that release NH₄⁺ from which NH₃ is evaporated). Therefore, the statement "the emission potential is unchanged" is not accurate. We change it to "the fertilizer application rate is unchanged", which results in the same soil content of NH₄⁺.

We changed the title of Section 5, from "**Ammonia under future scenarios**" to "**The effect of temperature change on the volatilization of ammonia**". To better reflect the motives of the work done in Section 5.

**Appendix**

The referencing of multiple equations in the appendix was wrong. I have corrected them, but please check and make sure my changes make sense. Finally, please also confirm that the

calculation of the first constant in (A-9) (2.75e9) is correct. I was unable to reproduce this value, but that could be an error at my end.

We went through the calculation again, we confirm that this constant is correct. This equation is adapted from Wichink Kruit's PhD thesis (2010), equation A1.4, on p.166.

**Minor edits**

All the minor comments listed below were addressed one by one. The comments that needed an explanation are also answered below (in blue).

Line 42: … amounted to

Line 43: …a very reactive base, and constitutes …

Line 47: … total ammonia gas is believed …

Line 60: … as shown in the Appendix.

Line 61: …of ammonia in the water in the soil is a function of soil acidity (pH) and temperature…

Line 62: …and controlled by the dissociation …

Line 63: …exists in the gas phase, and therefore Henry's law can be used to describe ….

Line 84: …during the 2003-2019 period …

Line 85: …(2022), leading to increased volatilized ammonia, (due to increase in both nitrogen…

Line 87/88: … Between the years 2008 and 2018, the … columns is estimated to be …

Line 107: … ammonia to provide regional …

Line 116: where

Line 120: …c' is 100

Line 123: … It is a function of the roughness length …

Line 125: The sentence starting with "It can be explained by …" is unclear. Are the authors stating that a resistance model is used to calculate k?

The sentence was changed to clarify the meaning: "It can also be calculated using a resistance model, often used to explain the exchange between the surface and the atmosphere (Wentworth et al., 2014)".

Line 177: … in areas where fertilizers are applied.

Show the emission potential where?

The sentence was changed to clarify the meaning: In addition to croplands, in this study we show the emission potential in forests and grasslands/shrublands for comparison with values in the literature.

Line 204 : … we use the ECMWF European Earth Consortium climate model …

Lines 216-219:… two scenarios: the SSPP2-4.5, a "middle of the road" socio-economic scenario with a nominal 4.5W/m2 radiative forcing level by 2100, similar to the RCP-4.5 scenario, and

the SSP5-8.5, the upper edge of the SSP scenario spectrum with a high fossil-fuel development use the 21st century.

Line 238: …2011), which marks …

Lines 239-240: …The differences are likely due to sampling issues: only cloud-free data are used to retrieve ammonia and different sampling times: IASI ….

Line 248: … Therefore, assuming that meteorological …

The sentence was changed to: "Therefore, assuming that meteorological and soil parameters affecting one dataset (e.g. IASI NH$_3$) are applicable to the other (e.g. model simulation), this is known as the steady-state approximation."

Line 259: What does [not shown here] mean?

It means that we do not show it in this study, but we did the work.

Line 275: The lifetime of ammonia () is shown in Figure 2d.

Line 285: … air stagnation in that area

Line 287: … and (AQEG, 2012), and these PM2.5 particles can dissociate, releasing ammonia

Line 295: … considered the loss …

Line 296: adopted here …

Line 333: … literature. Note that ammonia transfer coefficients are not available for all land types.

Line 335: … in grey in Figure 4

Line 336: … and swine manure, therefore, this value was assigned to croplands…

Line 343: The sentence starting with "These values" should maybe be rewritten as :

These values obtained by using MODIS land cover types and published estimates of k values represent our best effort to realistic mass transfer coefficients, and therefore realistic soil emission potentials.

Line 348: Are the authors extrapolating or aggregating by averaging over each GEOS-Chem grid box? Please make this clear.

We are aggregating several k values and averaging them to match the resolution of GEOS-Chem. We changed the sentence in the manuscript as follows: "We then extrapolate aggregate the array with the k values from 500 m × 500 m to the resolution of GEOS-Chem (0.5° × 0.625° grid box)."

Lines 357-358: Maybe rewrite as: The $k$ value assigned for forests represents the SO2 exchange in high croplands; this value may be very different for ammonia, since NH3 can easily dissolve in the water film on leaves under conditions of high humidity.

Lines 360-363: Again I think the authors mean aggregate not extrapolate. Which several grids?

Isn't the MODIS grid just being aggregated to the GEOS-Chem grid?

The sentence is changed to clarify. Now it reads as: "While changing the resolution of a fine array (500 m × 500 m), several grid points are merged and averaged together in order to construct the coarser grid box (0.5° × 0.625°); the result is therefore an average that might mix croplands with neighboring forests/barelands/grasslands. This leads to a range of different $k$ values that are shown on Figure 4."

Line 365: Using a land type …
Line 392: What is the sentence starting with "Based upon …" supposed to convey?
This sentence was removed.

Line 399: … The emission potential does not agree in value with that of GEOS-Chem
The sentence was changed to: "In case 3, the emission potential agrees spatially with that of GEOS-Chem".

Line 407: … England, northern France, northeastern Spain and Poland….
Line 412: …potential with values ranging from ….
Line 413: Are croplands different from agricultural lands? If not, the sentence starting with "Our values" seems unnecessary.
Yes. This sentence was removed.

Lines 420-423: In this study, lower values than those measured in the field are expected. Therefore, we consider our results to be in good agreement with the values in Personne et al. (2015), since ours reflect a 31 day mean of an average of over a large area (55x70 km2).
Line 433: …soil content of …
Caption of Figure 6: … are explained in the discussion on Figure 5.
Line 485: …more severe in eastern Europe ….
Line 487: …up to +50%...

Line 505: … facing big challenges in air (??) or downwind of large agricultural regions ….
Missing words were added to the sentence, now it reads as: "[…] Europe might be facing big challenges in air quality for regions nearby or downwind agricultural regions, since chemistry and atmospheric transport (Figure 3) drive the loss of ammonia during the growing season in this part of the world."

Line 512: …where there is no ammonia …
**Appendix**
Line 569: where HNH3 is Henry's constant, which can be …
Line 580: Substituting Eq.(A-7) into (A-8) we get:
Line 586: Since in …
Line 591: … where FluxNH3 …

Line 597: .. Eq. (A-11) can be written as ..

Line 603: … using Eq. (A-9) in (A-13) we get:

**References**

Evangeliou, N., Balkanski, Y., Eckhardt, S., Cozic, A., Van Damme, M., Coheur, P.-F., Clarisse, L., Shephard, M. W., Cady-Pereira, K. E., & Hauglustaine, D. (2021). 10-year satellite-constrained fluxes of ammonia improve performance of chemistry transport models. *Atmospheric Chemistry and Physics*, *21*(6), 4431–4451. https://doi.org/10.5194/acp-21-4431-2021

Garrido-Perez, J. M., Ordóñez, C., García-Herrera, R., & Barriopedro, D. (2018). Air stagnation in Europe: Spatiotemporal variability and impact on air quality. *Science of The Total Environment*, *645*, 1238–1252. https://doi.org/10.1016/j.scitotenv.2018.07.238

Viatte, C., Abeed, R., Yamanouchi, S., Porter, W. C., Safieddine, S., Van Damme, M., Clarisse, L., Herrera, B., Grutter, M., Coheur, P.-F., Strong, K., & Clerbaux, C. (2022). $NH_3$ spatiotemporal variability over Paris, Mexico City, and Toronto, and its link to $PM_{2.5}$ during pollution events. *Atmospheric Chemistry and Physics*, *22*(19), 12907–12922. https://doi.org/10.5194/acp-22-12907-2022

Wichink Kruit, R. (2010). *Surface-atmosphere exchange of ammonia*.